



# Combinatorial observation ionospheric characteristics during tropical cyclone Debbie passing eastern Australia in 2017 using GPS and ion sounder

Fuyang Ke[1], Jinling Wang[2], Kehe Wang[3], Jiujing Xu[4], Yong Wang[5], Xinzhi Wang[1], Jian Deng[6]

[1]School of Remote Sensing and Geomatics Engineering, Nanjing University of Information Science & Technology, Nanjing 210044, China

[2]School of Civil and Environmental Engineering, The University of New South Wales, Sydney, 2025, Australia

[3]Space Weather Services, Bureau of Meteorology, Surry Hills, New South Wales, Australia

[4]School of Geodesy and Geomatics, Anhui University of Science and Technology, Huainan 232001, China

[5]Jiangsu Province Surveying and Mapping Engineering Institute, Nanjing 210013, China

[6]School of Computer and Information Engineering, Xiamen University of Technology, Xiamen 361024, China

*Correspondence to*: Fuyang Ke (ke.fuyang@qq.com)

**Abstract.** The ionospheric morphology responses to tropical cyclone passing over eastern Australia, named as DEBBIE in
2017, is investigated using Global Positioning System (GPS) Slant Total Electron Content (STEC), GPS ionospheric scintillation $S_4$ index and ionospheric characteristics by High Frequency (HF) radio. Based on the data analysis in this study, some significant morphological characteristics of ionospheric responses to tropical cyclone Debbie are identified as follows: a) As the GPS satellites PRN01 and PRN11 were passing above typhoon center, their ROTI (Rate of STEC index) is obviously increased. b) The $S_4$ intensity of the GPS ionospheric scintillations is significantly enhanced on March 27, which
mostly concentrate around tropical cyclone center and distribute over the region within 18°S-25°S. c) The stronger enhancement of $f_0F1$ and $f_0F2$ was observed by High Frequency at Townsville on March 28, when the distance between Townsville and the center of tropical cyclone Debbie was shorter. Regarding the coupling mechanism between the ionospheric disturbance and the tropical cyclone, it is suspected that the electric field perturbations due to turbulent top movement from tropical cyclones might generate ionospheric irregularity and disturbance. To a certain extent, some
ionospheric irregularities can further produce bubbles.

## 1. Introduction

It is acknowledged that the ionosphere is an important and indispensable part of atmosphere for earth. The significant characteristic of ionosphere is high variation and complex physical system. The ionospheric irregularity and disturbance subject to the exterior impact and influence can degrade the operation reliability of ground-based radio and GPS or even lead
to their failure due to loss of signal lock. Normally, major ionospheric disturbances are caused by the strong solar and





geomagnetic activity. It has been indicated that ionospheric dynamics behaviors at low, mid and high latitude are dominated by solar tides and horizontal geomagnetic field lines, the inner magnetosphere and neutral winds, and the solar wind and electron precipitation, respectively (Skone et al., 2001). On the other hand, the lower layer of the ionosphere is connected to the neutral atmosphere. In the 1950s, Beynon et al. (1953) pointed out that ionospheric disturbance is related to the atmospheric activity in the troposphere. Moreover, the lower-level atmospheric activity not only can cause the ionospheric structure and physical variation, but also trigger the small and medium scale ionospheric disturbances (Forbes, 1996).

The tropical cyclone is a typical and strong convective atmospheric activity. Bauer (1958) firstly discovered one phenomenon of ionospheric response to hurricane passage that the critical frequency $f_0F2$ in the F2 ionospheric layer observed by High Frequency Radio (HF) began to increase as hurricane approaching to HF observation station. Additionally, the total electron content (TEC) and ionospheric scintillation can be achieved by ground GPS station to study the morphological characteristic of ionospheric response to tropical cyclone (Yang and Liu, 2016). Some previous studies have shown that the increasing TEC anomalies were detected during typhoon Matsa in 2005 (Mao et al., 2010), Aili in 2004 (Cheng et al., 2013), Nakri in 2008 (Lin, 2012), Tembin in 2012 (Yang and Liu, 2016) and Typhoon Meranti in 2016 (Chou et al., 2017). However, another study demonstrated that the TEC around the equatorial area decreased during typhoon Mahasen in 2013 and Hudhud in 2014 passing over the Indian sector (Guha et al., 2016). Meanwhile, the previous results showed that the ionospheric disturbances were emanating outward and lasting for more than 10 h before Super Typhoon Meranti in 2016 landfall (Chou et al., 2017), and that the number of radio occultation scintillation arriving at peak as typhoon Tembin in 2012 closest to Hong Kong (Yang and Liu, 2016). Using HF and Very Low (VL) Doppler radar, the results of statistical analysis showed that ionospheric disturbance percentage of 24 strong typhoons in China (Xiao et al., 2007), 41 tropical cyclones in the Atlantic Ocean (Nina et al., 2017), and 25 hurricanes in western and central part of the Czech Republic were 92%, 88% and 8%, respectively. Moreover, the characteristics of ionospheric disturbance intense were not the same. The results (Xiao et al., 2007; Rice et al., 2012; Yu et al., 2010; Ke et al., 2018) showed that the $f_0F2$ was increased during studied cyclones, but the studies (Liu et al., 2006; Rozhnoi et al., 2014) demonstrated that it is decreased. Using the sounder instruments on board the Cosmos 1809 satellite (Isaev et al., 2010), it is indicated that the pressure of the electron gas, electric field and scintillation intense increased in some specific zones, and plasma density and pressure above typhoon eye sharply decreased during typhoon intensification.

Thus, there are uncertainty and some controversies about morphological characteristics of ionospheric disturbances caused by tropical cyclones (Perevalova et al., 2011; Zakharov & Kunitsyn, 2012). Moreover, though there were also much many tropical cyclones in the southern hemisphere, rare tropical cyclones were used as cases to study the ionospheric response to them. Therefore, the previous studies were still limited not only in ionospheric observation instrument but also in the representative of the tropical cyclone cases studied worldwide. In the southern hemisphere, there are several ion sounder radio and GPS continuously operating reference stations (CORS) distributed around Australia for detecting the ionospheric morphological parameter. In March 2017, tropical cyclone Debbie was the strongest tropical cyclone in the Australian region since Cyclone Quang in 2015, which was branded the most dangerous cyclone to impact Queensland since Cyclone



Yasi in 2011. The combination observation of HF Doppler radar, GPS STEC and GPS ionospheric scintillation will be utilized to study the morphological characteristics of ionospheric response to tropical cyclone Debbie. This study will be a representative case of the ionospheric response to tropical cyclone in the southern hemisphere. It is also valuable to real the coupling mechanism between ionospheric disturbances and tropical cyclones.

In the following section, we will introduce dataset and methodology used in this study. Then, the ionospheric response to

cyclone Debbie in 2017 will be analyzed for demonstrating the morphological characteristics of ionospheric response to tropical cyclone Debbie in the southern hemisphere. Meanwhile, the possible coupling mechanism of reaction between ionospheric disturbance and tropical cyclone will be discussed.

## 2. Dataset

### 2.1 Tropical cyclone Debbie and ionospheric dataset

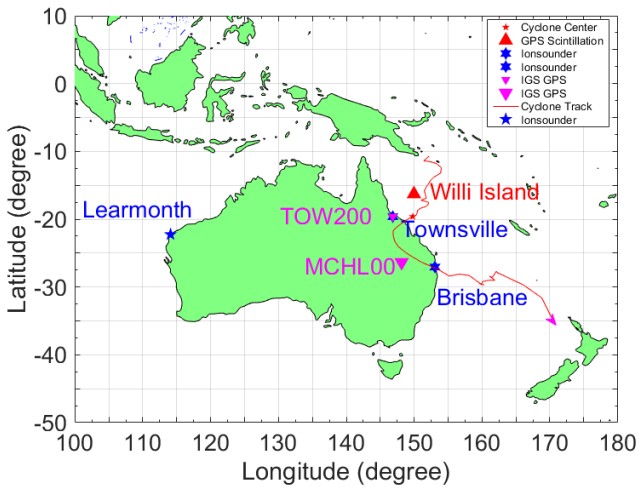


**Figure 1: GPS stations of ISMs (Red triangles: Willi Island), Ionosonde stations (Blue triangles, L: Learmonth, T: Townsville, B: Brisbane), paths of tropical cyclone DEBBIE (Red line), the tropical cyclone moving directions (arrows) and the places with the largest wind velocity (Red pentagrams)**

Tropical cyclone Debbie was formed above the south of Solomon Sea of South Pacific at Universal Time (UT) 12:00 of 21

Mar 2017. Then it landed on Hook Island of Queensland at UT00:00 on 28 Mar 2017 at the wind speed of 105m/s. Tropical cyclone Debbie left the Australian continent from Brisbane at 12:00 on 30 Mar 2017. As shown in Figure 1, tropical cyclone Debbie moved from north to south and its impact zone was in medium geomagnetic latitude from $15°$S to $60°$S in the southern hemisphere. Tropical cyclone Debbie center passed above Willi GPS ionospheric scintillation station for $S_4$ index, Townsville and Brisbane HF Doppler Radar stations for ionospheric parameters operated by Ionospheric Prediction Services

(IPS) Radio and Space Weather Services (SWS) of Australia, TOW2 and MCHL IGS GPS stations for GPS STEC belonging to IGS Network. When tropical cyclone Debbie landed on Hook Island of Queensland at UT00:00 on 28 Mar 2017 at the





wind speed of 105 m/s, the ellipsoidal distances between the tropical cyclone center and Willis GPS station, Townsville IGS, MCHL IGS, Townsville ion sounder station, Brisbane ion sounder station were 460 km, 230 km, 670 km, 230 km and 925 km, respectively. The shortest distances from the tropical cyclone center to these stations above were 160 km at UT00:00 of 24 March 2017, 230 km at UT00:00 of 28 March 2017, 167 km at UT12:00 of 29 March 2017, 230 km at UT00:00 of 28 March 2017 and 925 km at UT12:00 of 30 March 2017, respectively.

## 2.2 Solar and geomagnetic field activity

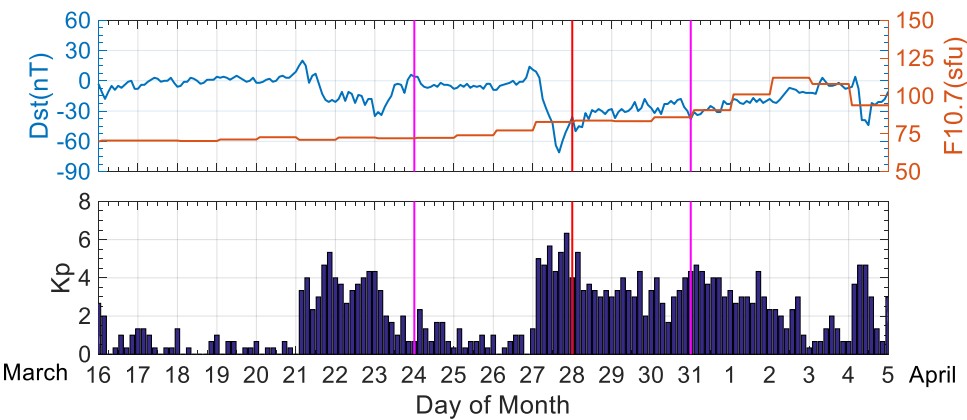

**Figure 2: The solar index F10.7, the index Dst and Kp of geomagnetic field during the period from16 March to 5 April 2017**

The ionospheric activity is mainly dominated and affected by the solar and geomagnetic activities. Therefore, the influence from the solar and geomagnetic field should be firstly analyzed before studying the ionospheric response to tropical cyclone. Normally, the solar radio flux (10.7cm/2800MHz, F10.7), geomagnetic activity indexes Dst and Kp were used to judge the solar and geomagnetic activity level, respectively. Their variations during tropical cyclone Debbie are shown in Figure 2. Generally, the F10.7 ranges of [70sfu, 100sfu], [100sfu, 150sfu], and [150sfu, 250sfu] represent low, moderate, and high level of solar activity, respectively [Wang et al., 2015]. The Dst ranges of $-50\,\mathrm{nT} < \mathrm{Dst} \leq -30\,\mathrm{nT}$, $-100\,\mathrm{nT} < \mathrm{Dst} \leq -50\,\mathrm{nT}$, $-200\,\mathrm{nT} < \mathrm{Dst} \leq -100\,\mathrm{nT}$ and $\mathrm{Dst} \leq -200\,\mathrm{nT}$ signify small, moderate, large and stronger and severe geomagnetic storms, respectively [Mao et al., 2010]. The Kp index ranges of 0-1, 2-4, 5, 6, and 7-9 denote quiet geomagnetic field, unstable geomagnetic field, small geomagnetic storm, large geomagnetic storm and severe geomagnetic storm, respectively. It is shown that the solar activity during tropical cyclone Debbie was at low level with F10.7 less than 100 sfu. But the Dst and Kp indexes indicate there were some a small geomagnetic storm on 27 March 2017 before cyclone Debbie landed on Hook Island of Queensland at UT00:00 on 28 Mar 2017. Therefore, the influence on ionosphere from the small geomagnetic storm should not be ignored to studying the ionospheric response to tropical cyclone Debbie.





## 3 Methodology

### 2.1 The Rate of GPS STEC

The rate of GPS slant TEC (ROT) is as a measure of GPS phase fluctuation, which can be used to monitor the ionospheric irregularity and disturbance. It represents the derivative of GPS slant TEC between two successive epochs, which can be calculated by the following equation (Yang and Liu, 2016):

$$\text{ROT} = \frac{\text{STEC}_k^i - \text{STEC}_{k-1}^i}{t_k - t_{k-1}} \tag{1}$$

where $i$ is the No of a GPS satellite; $t_k$ is an epoch time; the unit of ROT is TECU/min.

Rate of STEC index (ROTI) indicates the extent of the GPS phase fluctuations, which can be used to detect the occurrence of the ionospheric irregularities by the sharp TEC gradient. The ROTI represents the standard deviation of the ROT in a specific time interval (Yang and Liu, 2016):

$$\text{ROTI} = \sqrt{\langle \text{ROT}^2 \rangle - \langle \text{ROT} \rangle^2} \tag{2}$$

where the angle brackets denote the average taking over a 5-min time interval. These indices are calculated for each visible GPS satellite over a ground-based GPS station.

### 2.2 GPS ionospheric Scintillation

The intense of GPS ionospheric scintillation is typically quantified by the $S_4$ index, which is calculated using the following equation (Kintner et al., 2007):

$$S_4 = \sqrt{\frac{\langle I^2 \rangle - \langle I \rangle^2}{\langle I \rangle^2} - \sqrt{\frac{100}{\overline{\frac{C}{N_0}}} \left[ 1 + \frac{500}{19 \overline{\frac{C}{N_0}}} \right]}} \tag{3}$$

where $I$ is the intense of GPS signal, which is output from the GPS receiver tracking loop. $\overline{C/N_0}$ is the average Signal-to-Noise Ratio of satellite L1 Band in an observation period. If $S_4 > 0.2$, it indicates strong GPS ionospheric scintillation (Muella et al., 2008).

### 2.3 Ionospheric characteristics by HF radio

The ionospheric parameters of $f_0$E, $f_0$F1, and $f_0$F2 can be used to analyze the ionospheric response to cyclone in vertical direction. Accordingly, the electron density $N_e$ in each layer can be calculated using the frequency of reflection radio wave by the follow equation:

$$N_e = 1.24 \times 10^{10} f^2 \tag{4}$$






where $f$ is the critical frequency $f_0E$, $f_0F1$ and $f_0F2$ for the corresponding plasma with electron density $N_e$. The units of f and $N_e$ are MHz and $m^{-3}$, respectively. The electron density $N_e$ is linear dependent on $f^2$. Therefore, the parameters of $f_0E$, $f_0F1$, $f_0F2$ can reflect the electron density variation of ionosphere in vertical layers in response to tropical cyclone.

## 4. Analysis Result and Discussion

**4.1 GPS STEC response to tropical cyclone Debbie**

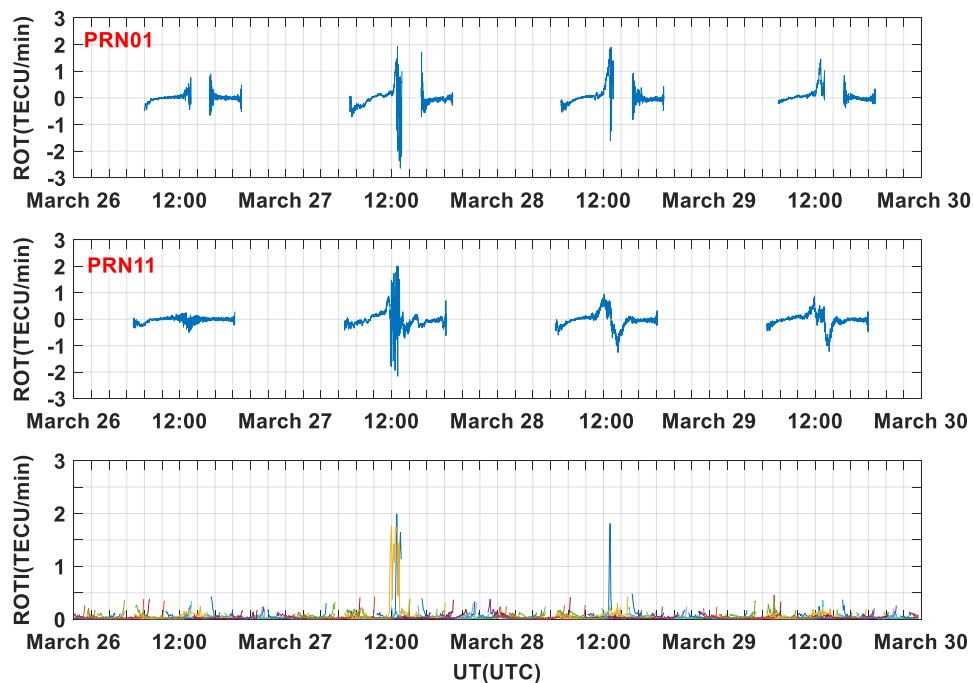

**Figure 3: The variation of ROT for GPS PRN01 and PRN11 and ROTI (5min) for all satellites during March 26-29 over the TOW2 IGS station.**

The GPS STEC response to passing tropical cyclone Debbie from 26-29 March 2017 is firstly analyzed. The GPS STEC was
extracted from TOW2 IGS station, which tropical cyclone Debbie passed above. The variation of ROT for PRN01 and PRN11 and ROTI (5min) for all satellites during March 26-29 are shown in figure 3. It shows that ROTI has an evident increment on UT 12:00 (Local Time (LT) =UT+8h) March 27, the day before tropical cyclone Debbie center landed on Hook Island. When tropical cyclone Debbie center landed on Hook Island of Queensland on March 28, the increasing of ROTI became slight. As shown in the top two of Figure 3, the variation of ROT for PRN01 and PRN11 were obviously
increasing as the cyclone Debbie landing. The IPP traces of GPS PRN01 and PRN11 satellites over TOW2 station were above the impact area of tropical cyclone Debbie on 28 March. Although there was a small geomagnetic storm on 27 March,





GPS STEC extracted by other GPS satellites were not anomalous. Therefore, it can be inferred that the anomaly of ROTI and ROT extracted by GPS PRN01 and PRN11 above tropical cyclone center on UT12:00 of 27 and 28 March were more likely triggered by cyclone Debbie.

**4.2 GPS ionospheric scintillation response**

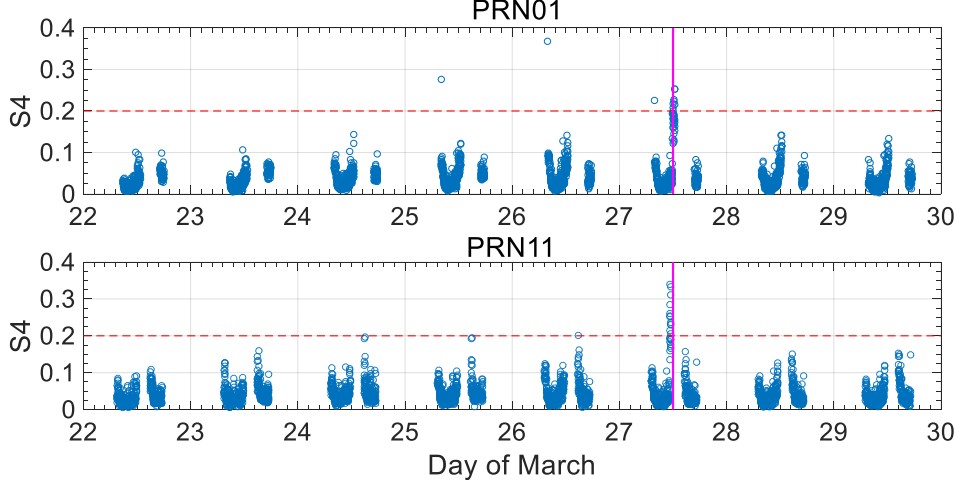

**Figure 4: The GPS ionospheric scintillation S4 variations of GPS PRN01 and PRN11 satellites during 22-29 March 2017. The dotted red line is the threshold of the strong GPS ionospheric scintillation. The magenta vertical line denotes the time point when cyclone Debbie center was the closest to GPS station**

Although ionospheric scintillation can degrade the GPS signal quality or even cause failure of the signal lock, GPS also provides a new tool for detecting ionospheric irregularity and scintillation. GPS scintillation amplitude index $S_4$ more than 0.2 indicates strong ionospheric scintillation. Figure 4 shows that the variation of GPS ionospheric scintillation of PRN01 and PRN11 above Willi Island GPS station during the period of 22-29 March 2017. When the distance from tropical cyclone center to Willis station was 370 km and the center wind speed was 105 m/s at midnight of 27 March, the number and

intensity of S4>0.2 observed by GPS satellite G01 and G11 were more and larger than those at the other time. It shows that the geomagnetic activity was with a small storm in Figure 2. Under the geomagnetic condition, the anomaly of ionospheric scintillation of GPS PRN01 and PRN11 might be related to the small geomagnetic storm. But the $S_4$ values of the other GPS satellites are all smaller than the $S_4$ threshold. Hence, it can be deduced that the number increment of GPS ionospheric scintillations might be triggered by tropical cyclone Debbie on March 27.



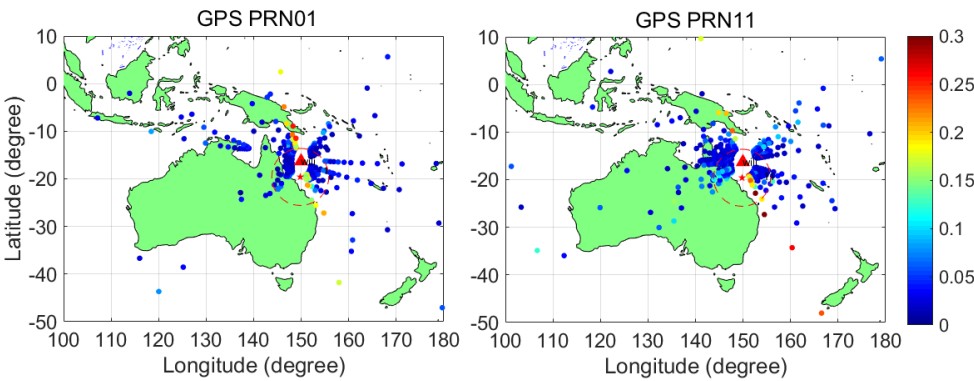

**Figure 5: Ionospheric puncturing point traces and GPS ionospheric scintillation $S_4$ intensity of GPS PRN01 and PRN11 satellites through cyclone Debbie on March 27. The red dashed cycles indicate the area affected by cyclone Debbie. The colorful solid circles are GPS ionospheric scintillations and their intensity**

To further determine the correlation between GPS ionospheric scintillation and tropical cyclone Debbie, the spatial relations between GPS ionospheric scintillation points puncturing into ionosphere and tropical cyclone Debbie are shown in Figure 5. It is obvious that more points of $S_4>0.2$ are mainly distributed around the tropical cyclone center. Moreover, the intensity and number of the points of $S_4>0.2$ above the area of $18°S - 25°S$ in the latitude and $150°E - 155°E$ in the longitude around tropical cyclone center ($B = 19.6°S, L = 149.8°E$) was stronger and larger than those above the other area. Normally, the occurrence of strong ionospheric scintillations is more frequent in low geomagnetic latitude ($\pm15°$) and high geomagnetic latitude ($\pm70°$). However, the area of these GPS ionospheric scintillations with $S_4>0.2$ doesn't belong to the area of frequent ionospheric scintillation. The evidence further verified that the strong ionospheric scintillations above the area around tropical cyclone Debbie might be triggered by tropical cyclone Debbie.

**4.3 The ionospheric parameters in E/F1/F2 layer by ion sounder**

The ionosphere is divided into D, E, F1 and F2 layers. There are differences in the characteristics of the ionosphere in each layer. However, it is difficult to distinguish the characteristics of the ionosphere in each layer in response to tropical cyclone Debbie only by GPS. Therefore, HF radios installed in Townsville and Brisbane are used to detect and analyze the characteristic parameters of $f_0E$, $f_0F1$ and $f_0F2$ in E, F1 and F2 layer for ionospheric response to cyclone Debbie. With regard to characteristic parameters $f_0E$, $f_0F1$ and $f_0F2$, the best fits for their mean value from 24 to 31 March were referred as their normal values to judge electron density response to tropical cyclone in vertical layers.



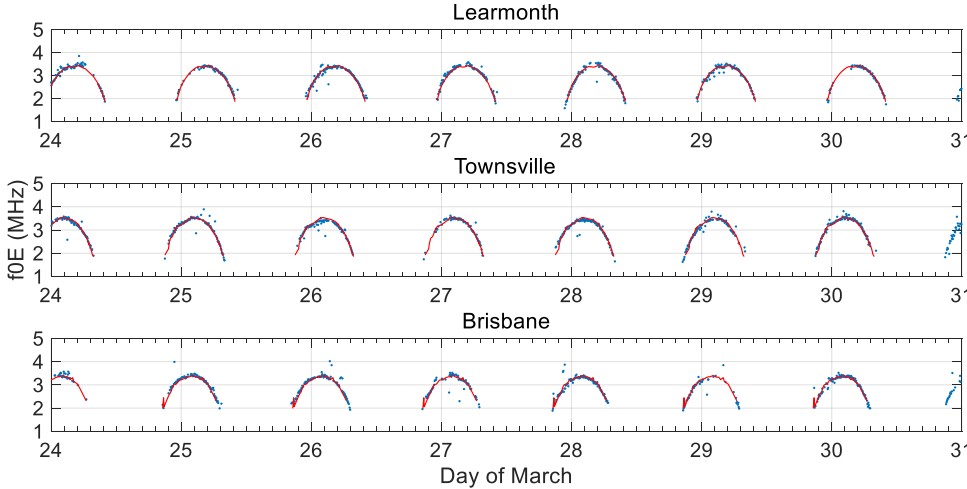

**Figure 6: The $f_0E$ variation in the ionospheric E layer at Learmonth, Townsville and Brisbane HF radar during the period of 24 to 30 March in 2017 as tropical cyclone Debbie moving. The $f_0E$ is the ordinary wave critical frequency of the lowest thick layer which causes a discontinuity. The red lines denote the best fit for the mean value of $f_0E$ from 24 to 30 March**

The characteristic parameters $f_0E$ in the ionospheric E layer from Townsville and Brisbane HF radio stations as tropical cyclone Debbie moving from 24 to 30 March are shown in Figure 6. Although $f_0E$ values of Townsville were small enhanced and elevated at midday of 29 and 30 March after tropical cyclone Debbie landfall, the $f_0E$ values of Learmonth, Townsville and Brisbane on landfall day and other days were no significant anomaly relatively to their reference values. The $f_0E$ of Learmonth, Townsville and Brisbane were all approximately equal. The ranges of $f_0E$ were respectively from 1.7 to 4.0 MHz and from 85 km to 100 km. The phenomenon in the ionospheric E layer indicates that the ionosphere in E layer is not disturbed by tropical cyclone Debbie.

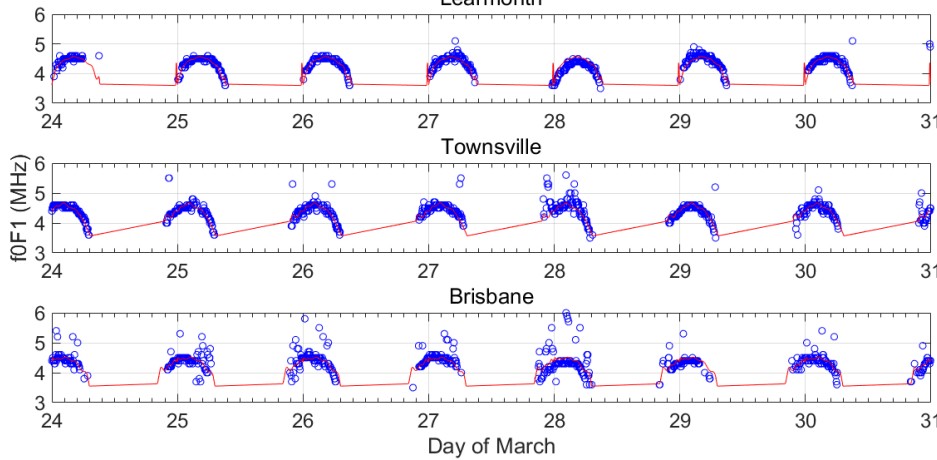

**Figure 7: The $f_0F1$ variation in the ionospheric F1 layer at Learmonth, Townsville and Brisbane HF radar stations during the period from 24 to 30 March in 2017 as tropical cyclone Debbie moving. The $f_0F1$ is the ordinary wave F1 critical frequency. The red lines denote the best fit for the mean value of $f_0F1$ from 24 to 30 March**





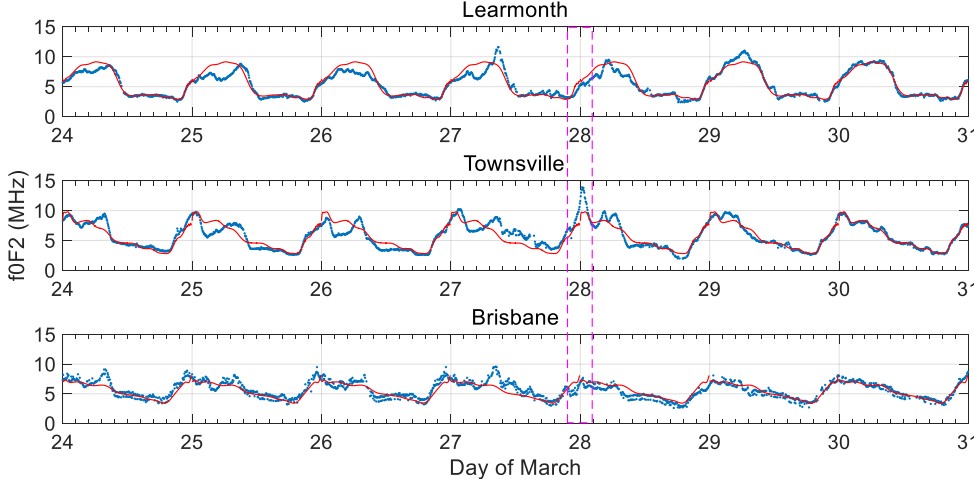


**Figure 8: The $f_0$F2 variation in the ionospheric F2 layer at Learmonth, Townsville and Brisbane HF radar stations during the period of 24 to 30 March in 2017 as tropical cyclone Debbie moving. The $f_0$F2 is ordinary wave critical frequency of the highest stratification in the F region. The red lines denote the best fit for the mean value of $f_0$F2 values from 24 to 30 March**

The ionospheric parameters $f_0$F1 and $f_0$F2 on Learmonth, Townsville and Brisbane HF radio stations as tropical cyclone

Debbie moving from 24 to 30 March are shown in Figure 7 and Figure 8. It is obvious that the $f_0$F1 and $f_0$F2 intensity on Townsville station on 28 March were significantly stronger than the best fit value. However, the $f_0$F1 and $f_0$F2 intensity on Learmonth and Brisbane are approximately equal to the best fit values under the same solar and geomagnetic activity. Theoretically, the influences on $f_0$F1 and $f_0$F2 of Learmonth and Townsville in the same latitude from the small geomagnetic storm on 27 and 27 March should be almost the same. However, the anomaly extent of $f_0$F1 and $f_0$F2 on Townsville station

are significantly larger than those on Learmonth on 28 March. The largest $f_0$F1 deviation on Townsville station relative to their best fits was 1.0MHz - 1.5MHz. Additionally, the largest $f_0$F2 deviation on Townsville station relative to their best fits was 5.0MHz. The electron density $N_e$ is linear related to $f_0$F1 and $f_0$F2. Thus, it can be inferred that the electron density of F1 and F2 significantly increased on the tropical cyclone Debbie landfall day. Moreover, the enhancement $f_0$F1 and $f_0$F2 on Townsville station on landfall day were stronger than those of Brisbane, when the ellipsoidal distances from tropical cyclone

center to Townsville and Brisbane were 230 km and 925 km, respectively. Therefore, the stronger $f_0$F1 and $f_0$F2 enhancement of Townsville station on 28 March should be attributed to tropical cyclone Debbie.

**4.3 The mechanism of ionospheric response to tropical cyclone**

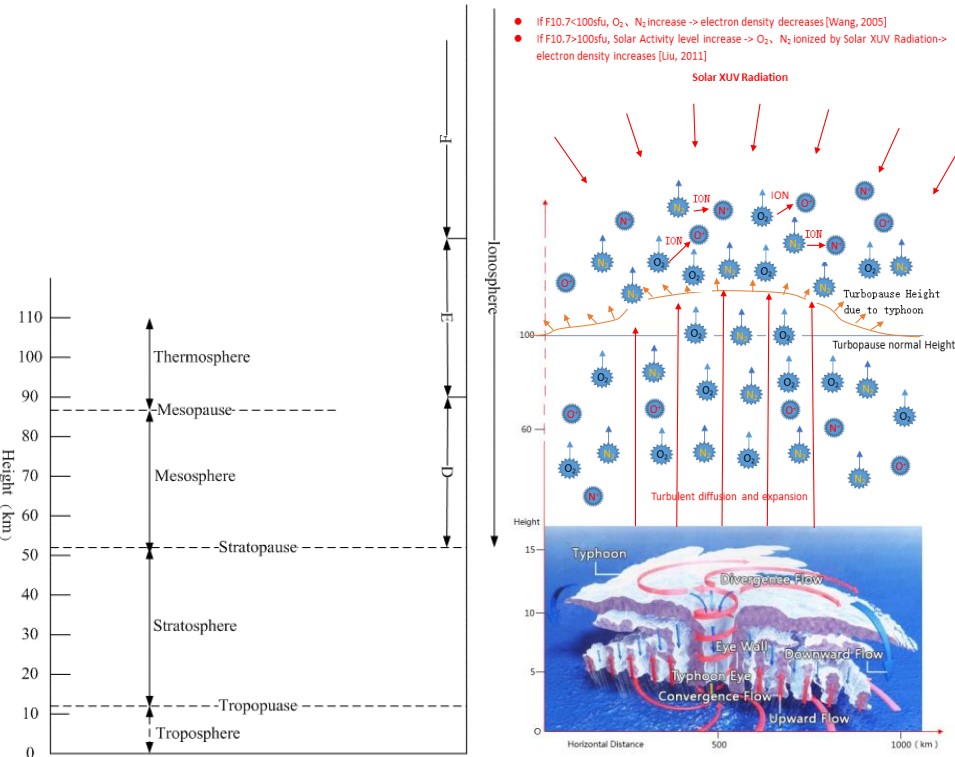

**Figure 9: Profile of atmosphere and tropical cyclone structure (From China Meteorological Administration http://www.cma.gov.cn) and the schematic diagram of ionospheric response to tropical cyclone**

The above results reveal that tropical cyclone Debbie can cause ionospheric irregularity and disturbance, which can further produce ionospheric scintillations. However, the coupling mechanism between tropical cyclone and ionosphere disturbances is still indefinite (Shen, 1982). So far, there are following proposed coupling mechanisms, such as gravity waves generated by tropical cyclone (Xiao, et al. 2007), an atmosphere divergence/convergence model and dynamic coupling (Shen, 1982), a disturbed electric field (Isaev, et al. 2002), turbulent top moving (Shen, 1982; Liu, et al. 2006; Wang, et al. 2005) and lighting discharge (Shao, et al. 2013). According to the morphological characteristics of ionospheric response to tropical cyclone Debbie, the coupling mechanism will be discussed based on turbulent top moving, the structure of atmosphere and typhoon (Figure 9), and electron photochemical reactions.

Tropical cyclone belongs to airflow system in troposphere with a height of 20 km or less, which might not directly affect the ionosphere with a height of 50 km or more in the left of Figure 9 (Shen, 1982). But the strong airflow of lower layer in troposphere can lead to the structure change of stratosphere and mesosphere (Shen, 1982; Liu, et al. 2006). The airflow direction of the tropopause around tropical cyclone center in the right of Figure 9 is downward. The convergence of airflow will increase the height of the stratosphere upward above tropical cyclone (Liu, et al. 2006). Then the upward airflow will continue to develop upward due to the temperature structure of the middle layer and elevate the height of the turbulence layer, whose height is about 100km (Shen, 1982). According to atmospheric turbulence layer movement theory (Liu, et al.



2006), it will make the turbulent diffusion coefficient increase and the molecular diffusion coefficient decrease. As a result, several neutral molecules ($N_2$, $O_2$) in E layer will be taken into the ionospheric F1 and F2 layer. X-rays and extreme ultraviolet rays from the sun, which are absorbed by the upper atmosphere, can make these neutral molecules to ionize and produce many free electrons and ions leading to the increment of electron density in F1 and F2 layer (Liu, et al. 2011). Therefore, it can be explained that the phenomenon that the $f_0$F1 and $f_0$F2 on Townsville station were significantly enhanced as tropical cyclone Debbie landed on and is the closest to Townsville station. At the same time, the growth of Rayleigh-Taylor instability due to the electric field perturbations can produced ionospheric disturbance in the F layer (Prakash 1999). The ionospheric density irregularities can form drift waves (Kelley 1985), which can develop and rise into the topside F region ionosphere to create bubbles. When GPS signals encounter the bubbles, GPS ionospheric scintillations will occur.

## 5. Conclusions

The morphological characteristics of ionospheric response to tropical cyclone Debbie passing eastern Australia in 2017 is investigated by GPS and ion sounder. The results agree with the previous viewpoint that tropical cyclone can trigger ionospheric disturbance. The morphological characteristics of ionospheric response to tropical cyclone Debbie can be summarized as follows.

(1) As the GPS satellites PRN01 and PRN11 were passing over tropical cyclone Debbie, their ROTI and ROT are significantly increased. (2) The $S_4$ intensity of GPS ionospheric scintillations is enhanced on March 27, which mostly concentrate above tropical cyclone center and distribute over the region of $18^\circ S - 25^\circ S$ in the latitude and $150^\circ E - 155^\circ E$ in the longitude around tropical cyclone center (B $= 19.6^\circ S, L = 149.8^\circ E$)。(3) Compared with those on Learmonth and Brisbane, the intensity of $f_0$F1 and $f_0$F2 on Townsville was obviously increased as tropical cyclone Debbie landed on 28 March. At the same time, distance between Townsville and the tropical cyclone center is the shortest.

Considering the influence from the geomagnetic and solar activity, the turbulent top movement theory is utilized to explain how tropical cyclone causes ionospheric irregularity and further triggers ionospheric scintillations. It is assumed that the turbulent top movement of tropical cyclone can break the balance of the electric field. Then, the electric field perturbations can contribute to the growth of Rayleigh-Taylor instability producing ionospheric disturbance in the F layer of the ionosphere. Furthermore, the ionospheric density irregularities can form drift waves to create bubbles in F layer. When GPS signals encounter the bubbles, GPS ionospheric scintillations can appear.

## Funding

This study is supported by the National Natural Science Foundation of China (grant no. 41674036 and 41704008), Key Research & Development Program of Jiangsu (grant no. BE2016020), QingLan Project of Jiangsua.





**Acknowledgements**

We acknowledge International GPS Service for GPS raw data and the Bureau of Meteorology of the Australian Government for providing us with the tropical cyclone, GPS ionospheric scintillation and ionosonde data sets. Finally, we are grateful to the US National Oceanic and Atmospheric Administration and Geomagnetic Data Center for the F10.7, Kp and Dst data support.

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
