# Peer review of "Combinatorial observation ionospheric characteristics during tropical cyclone Debbie passing eastern Australia in 2017 using GPS and ion sounder"

_Annales Geophysicae, 2019_

## Referee Comment (RC1) · Anonymous Referee #1 · 2 Aug 2019

The authors reported a multi-instrument experiment to study the effects of tropospheric cyclone on overlying ionosphere. The topic corresponds to the profile of the journal. The topic is relevant since there are still many open questions in connection with the troposphere - upper atmosphere/ionosphere coupling mechanisms. The present study investigated the ionospheric response to tropical cyclone named DEBBIE passing over eastern Australia using GPS and ionosonde data. The processes within the ionosphere was very complex, since the investigated meteorological event occurred during the main and recovery phase of a moderate geomagnetic storm. Therefore, this is

an interesting event to investigate its impact on the ionosphere. However, the authors needs to complete the introduction part and deepen the physical discussions of the observed results. Therefore, the authors need to improve significant modifications. Furthermore, the authors need to improve English significantly. Thus, I suggest a major revision before the acceptance of the manuscript to publish. General comments: 1.) The authors called in different strange names the ionosondes: e.g. "The stronger enhancement of f0F1 and f0F2 was observed by High Frequency at" and "ion sounder" in the title, abstract and in the whole text. I suggest to use the further expressions to name this instrument: High Frequency radio sounder or ionosonde. 2.) The English of the whole text have to be improved. Please, use the same, past or present tense during the whole manuscript. Take care of the word order of the sentences, check singular/plural, furthermore the prepositions. Avoid the repetition of a word. 3.) Introduction: In the discussion part the author count the different coupling mechanisms but they didn't detail them in the introduction. Please, write 1-2 paragraphs about the different mechanisms what you listed in the discussion with more reference. 4.) Page 2 line 50: I miss a reference in connection with the hurricanes in Czech republic: "the results of statistical analysis showed that ionospheric disturbance percentage of 24 strong typhoons in China (Xiao et al., 2007), 41 tropical cyclones in the Atlantic Ocean (Nina et al., 2017), and 25 hurricanes in western and central part of the 50 Czech Republic were 92%, 88% and 8%, respectively" 5.) Page 3 Fig. 1 and its caption: Why do you indicate the same type of symbol more times on the Figure? Caption: I only see blue stars and one arrow and one red pentagram on the Figure. Please, correct the caption. 6.) Page 3 line 80 and more times after that: "the wind speed of 105m/s" Are you sure in this data? 105 m/s = 378 km/h! 7.) Page 4 line 89-91: This sentence is a little bit confusing: "The shortest distances from the tropical cyclone center to these stations above were 160 km at UT00:00 of 24 March 2017, 230 km at UT00:00 of 28 March 2017, 167 km at UT12:00 of 29 March 2017, 230 km at UT00:00 of 28 90 March 2017 and 925 km at UT12:00 of 30 March 2017, respectively." It would be better the define the distance station by station. 9.) Page 6 line 148 and Fig. 3.: What was the

exact time when Debbie landed on Hook Island? I can see more increase on 27 Mach also in ROT for PRN01 and PRN11. 10.) About the 4.1 and 4.2 subsections and Fig 3 and 4.: It would be nice to see a measurement of a reference satellite for away from the cyclone. 11.) Fig. 5.: I can see more red point north from the red circle but at the same longitude. Can you detail the reason of those red dots/anomalies? 12.) In connection with subsection 4.3: Why don't you use the monthly median or an average of reference days (geomagnetically quiet days) as reference and compare it with ionospheric parameters (foE, foF2, foF1) observed during the investigated period? If you use: "best fits for their mean value from 24 to 31 March . . . as their normal values" you use the parameters observed during geomagnetically disturbed period as reference. Thus I suggest to repeat the analysis using the monthly median of the parameters or detected during geomagnetically calm periods. 13.) Fig. 8. What does the pink rectangular good for? Please, detail it in the caption of the figure. 14.) Page 10. line 217: "The electron density ðÍŚĄðÍŠŠ is linear related to f0F1 and f0F2." It is not true. Look at the eq. (4) at the manuscript. 15.) Page 10. line 220: "Therefore, the stronger f0F1 and f0F2 enhancement of Townsville station on 28 March should be attributed to tropical cyclone Debbie". I can see strong foF1 variation also at Brisbane. How can you explain it? 16.) Discussion: The discussion is very short. In the first paragraph the authors count the different coupling mechanisms. But they didn't discuss them and didn't compare them with their findings. They say that the morphological characteristic of the ionospheric variations what they found agree only one mechanism of them (turbulent top movement) but they didn't explain why. Please, discuss you results more carefully and compare them with the findings of previous studies in the literature. Especially detail please the last 4 lines of the discussion, the mechanism in connection with the generation of the plasma bubbles.

Minor comments: - The title of 4.3 subsection: "4.3 The ionospheric parameters in E/F1/F2 layer by ion sounder" change it to e.g.: "4.3 The ionospheric parameters in E/F1/F2 layer detected by ionosondes" or "4.3 The ionospheric parameters in E/F1/F2 layer derived from ionograms" - Page 9 line 199: "The ranges of f0E were respectively

from 1.7 to 4.0 MHz and from 85 km to 100 km." I suggest to change to: "The ranges of f0E were respectively from 1.7 to 4.0 MHz and its virtual height (h'E) from 85 km to 100 km. "

Please also note the supplement to this comment:
https://www.ann-geophys-discuss.net/angeo-2019-72/angeo-2019-72-RC1-supplement.pdf

---

## Short Comment (SC1) · 7 Aug 2019

This paper can be accepted I suggested In the Section Introduction It is acknowledged that the ionosphere is an important and indispensable part of atmosphere for earth. Please cite some past papers (at lease one
* * *

---

## Author Comment (AC1) · 5 Sep 2019

Comment #01: The authors called in different strange names the ionosondes: e.g. "The stronger enhancement of f0F1 and f0F2 was observed by High Frequency at" and "ion sounder" in the title, abstract and in the whole text. I suggest to use the further expressions to name this instrument: High Frequency radio sounder or ionosonde. Response #01: All High Frequency and ion sounder in this paper have been replaced by ionosonde.

[Figure]

Comment #02: The English of the whole text have to be improved. Please, use the same, past or present tense during the whole manuscript. Take care of the word order of the sentences, check singular/plural, furthermore the prepositions. Avoid the repetition of a word. Response #02: The English of the manuscript has been improved. The same tense has been used in the whole manuscript. The word of the sentences, singular/plural, and the prepositions have been checked and revised. The repetition of a word has been avoided. The details are shown in the revised manuscript.

Comment #03: Introduction: In the discussion part the author counts the different coupling mechanisms but they didn't detail them in the introduction. Please, write 1-2 paragraphs about the different mechanisms what you listed in the discussion with more reference. Response #03: The following description has been added in the fourth paragraph on page 2 lines 67-75. Additionally, there are also some controversies about the coupling mechanism between tropical cyclones and ionospheric disturbance. Hung et al. (1978) of NASA in the United States have found the existence of gravity waves and mesoscale disturbances in the F layer of the ionosphere during the tornado eruption using ionosonde, and considered that the gravity wave may be the main source of ionospheric disturbance. It is suggested (Shen, 1982; Liu, et al. 2006; Wang, et al. 2005) that the turbopause motion is a possible mechanism for the interaction between the lower layers of the atmosphere and ionosphere. It is considered that the electric field disturbance in typhoon or hurricane is activated by the ionospheric disturbance caused by the current perturbation from the charged water droplets and aerosols transmitted upward (Isaev 2002, 2010). So, the study on tropical cyclone Debbie is also valuable to realize the coupling mechanism between ionospheric disturbances and tropical cyclones.

Comment #04: Page 2 line 50: I miss a reference in connection with the hurricanes in Czech republic: "the results of statistical analysis showed that ionospheric disturbance percentage of 24 strong typhoons in China (Xiao et al., 2007), 41 tropical cyclones in the Atlantic Ocean (Nina et al., 2017), and 25 hurricanes in western and central part of

the 50 Czech Republic were 92%, 88% and 8%, respectively" Response #04: "...and 25 hurricanes in western and central part of the 50 Czech Republic (Šindelářová et al., 2009) were 92%, 88% and 8%, respectively." The reference in connection with the hurricanes in Czech Republic has been added on page 2 line 50. The referee: Šindelářová T., Burešová D., Chum J.: Observations of acoustic-gravity waves in the ionosphere generated by severe tropospheric weather, Studia Geophysica et Geodaetica, 53(3): 403-418, 2009.

Comment #05: Page 3 Fig. 1 and its caption: Why do you indicate the same type of symbol more times on the Figure? Caption: I only see blue stars and one arrow and one red pentagram on the Figure. Please, correct the caption. Response #05: The legend and caption in Fig. 1 have been corrected as the following figure.

Comment #06: Page 3 line 80 and more times after that: "the wind speed of 105m/s" Are you sure in this data? 105 m/s = 378 km/h! Response #06: Cyclone Debbie landed on Hook Island of Queensland at UT00:00 on 28 Mar 2017 at the wind speed of 105 knots, about 54 m/s. Corrections have been made in the paper.

Comment #07: Page 4 line 89-91: This sentence is a little bit confusing: "The shortest distances from the tropical cyclone center to these stations above were 160 km at UT00:00 of 24 March 2017, 230 km at UT00:00 of 28 March 2017, 167 km at UT12:00 of 29 March 2017, 230 km at UT00:00 of 28 90 March 2017 and 925 km at UT12:00 of 30 March 2017, respectively." It would be better the define the distance station by station. Response #07: The sentence has been adjusted to the sentence of "When tropical cyclone Debbie landed on Hook Island of Queensland, the ellipsoidal distances between the tropical cyclone center and the observation stations of Willis, Townsville and Brisbane were 460 km, 230 km and 925 km, respectively."

Comment #08: Page 6 line 148 and Fig. 3.: What was the exact time when Debbie landed on Hook Island? I can see more increase on 27 Mach also in ROT for PRN01 and PRN11. Response #08: Cyclone Debbie landed on Hook Island of Queensland at

UT00:00 on 28 Mar 2017. The increment of ROT for PRN01 and PRN11 on 27 March might be caused by the small geomagnetic storm.

Comment #9: About the 4.1 and 4.2 subsections and Fig 3 and 4.: It would be nice to see a measurement of a reference satellite for away from the cyclone. Response #9: The measurement of PRN23 far away from the cyclone is selected as the reference in Fig 3 and 4.

Comment #10: Fig. 5.: I can see more red point north from the red circle but at the same longitude. Can you detail the reason of those red dots/anomalies? Response #10: The north from the red circle belongs the low geomagnetic latitude, where the ionsopheric scintillations are highly frequent. Therefore, the ionospheric scintillations are caused by the ionospheric irregularities near the equator.

Comment #11: In connection with subsection 4.3: Why don't you use the monthly median or an average of reference days (geomagnetically quiet days) as reference and compare it with ionospheric parameters (foE, foF2, foF1) observed during the investigated period? If you use: "best fits for their mean value from 24 to 31 March. . . as their normal values" you use the parameters observed during geomagnetically disturbed period as reference. Thus, I suggest to repeat the analysis using the monthly median of the parameters or detected during geomagnetically calm periods. Response #11: This paper uses the monthly median as a reference, but the expression is inappropriate. The expressions have been corrected.

Comment #12: Fig. 8. What does the pink rectangular good for? Please, detail it in the caption of the figure. Response #12: The pink rectangular is just to emphasize on the tropical cyclone Debbie landfall day.

Comment #13: Page 10. line 217:" The electron density is linear related to f0F1 and f0F2." It is not true. Look at the eq. (4) at the manuscript. Response #13: The sentence has been changed to "The electron density is positively related to f0F1 and f0F2".

Comment #14: Page 10. line 220: "Therefore, the stronger f0F1 and f0F2 enhancement of Townsville station on 28 March should be attributed to tropical cyclone Debbie". I can see strong foF1 variation also at Brisbane. How can you explain it? Response #14: Although the f0F1 in F1 layer observed by Brisbane station is also increased on 28 March in Figure 7, the periodic anomaly of f0F1 in those days might be due to Ionosonde noise. And the explanation has been added on page 8 lines 224-226.

Comment #15: Discussion: The discussion is very short. In the first paragraph the authors count the different coupling mechanisms. But they didn't discuss them and didn't compare them with their findings. They say that the morphological characteristic of the ionospheric variations what they found agree only one mechanism of them (turbulent top movement) but they didn't explain why. Please, discuss you results more carefully and compare them with the findings of previous studies in the literature. Especially detail please the last 4 lines of the discussion, the mechanism in connection with the generation of the plasma bubbles. Response #15: Compared with our findings in the study, the different coupling mechanism has been discussed. Compared our results and the previous findings, the ionospheric disturbance and scintillation are explained. The detail discussion is as following. The results reveal that the ionospheric irregularity and disturbance could be likely related to tropical cyclone Debbie. Even it can further produce GPS ionospheric scintillations. However, the coupling mechanism between tropical cyclone and ionospheric disturbances is still indefinite and controversial. The previous studies considered that the source of ionospheric disturbance might be from gravity waves generated by tropical cyclone (Xiao, et al. 2007), an atmosphere divergence/convergence model and dynamic coupling (Shen, 1982), a disturbed electric field caused by tropical cyclone (Isaev, et al. 2002), turbulent top layer movement of tropical cyclone (Shen, 1982; Liu, et al. 2006; Wang, et al. 2005) and lighting discharge from clouds of tropical cyclone (Shao, et al. 2013). Nevertheless, the vertical gravity waves could not disturb the more than 100 km height ionosphere in effective range due to its dozens of kilometres wavelength. Especially, it is much difficult to explain why f0F2 in more than 300 km height F2 ionospheric layer

observed by Townsville Ionosonde station is larger than the monthly median value as tropical cyclone Debbie was landing on 28 March in Figure 8. Assuming that gravity waves can cause ionospheric disturbances, the f0E in ionospheric E layer should be also anomalous. On the contrary, the f0E is approximately equal to the monthly median value in Figure 6. It is also difficult to explain the phenomenon only using the turbulent top layer movement of tropical cyclone and the disturbed electric field above the tropical cyclone. Because the tropical cyclone belongs to airflow system in troposphere with a height of 20 km or less and might not directly affect the ionosphere with a height of 50 km or more in the left of Figure 9. Furthermore, the short-term lightning could not explain the long-term ionospheric disturbances for about 2 hours in Figure 8. Therefore, it is supposed that the ionospheric disturbance in response to tropical cyclone Debbie is interacted by multi-source of turbulent top layer movement, electric field and electron photochemical reactions. The strong tropical cyclone airflows can lead to the structure change of stratosphere and mesosphere. Among them, the upward airflow will continue to develop upward due to the temperature structure of the middle layer and elevate the turbulence layer with about 100km height (Shen, 1982). By contrary, the airflow direction of the tropopause above tropical cyclone centre is downward. According to atmospheric turbulence layer movement theory (Liu, et al. 2006), the airflows from the tropical cyclone will make the turbulent diffusion coefficient increase and the molecular diffusion coefficient decrease. As a result, some neutral molecules ($N_2$, $O_2$) in E layer will be taken into the ionospheric F1 and F2 layer. X-rays and extreme ultraviolet rays from the sun can make these neutral molecules to ionize and produce free electrons and ions leading to the increment of electron density in F1 and F2 layer (Liu, et al. 2011). Therefore, it can be explained the phenomenon that the f0F1 and f0F2 on Townsville station are significantly enhanced as tropical cyclone Debbie is above Townsville station in Figure 7 and Figure 8. Along with the increment of electrons, the balance of electric field is destroyed. The growth of Rayleigh–Taylor instability from the electric field perturbations can lead to some ionospheric irregularities in the F layer (Prakash 1999), which may have velocity shear

mixing within the hole gradients (Kelley 1985). When the hole arrives at the topside of F layer, the bubble is produced. As it happens that GPS signal encounters the bubble, ionospheric scintillation will occur. Therefore, it can explain why there are just some ionospheric scintillations of PRN01 and PRN11 above tropical cyclone Debbie, whose values of S4 are more than 0.2 in Figure 5. Moreover, the S4 of ionospheric scintillations of PRN01 and PRN11 observed by Townsville station is more than 0.2 on the midday of 27 March 2017 in Figure 4. Meanwhile, the f0F2 in ionospheric F2 layer is enlarged at the same time in Figure 8. Thus it can be supposed that the ionospheric scintillation is produced by the ionospheric irregularities in F2 layer due to tropical cyclone Debbie.

Please also note the supplement to this comment:
https://www.ann-geophys-discuss.net/angeo-2019-72/angeo-2019-72-AC1-supplement.pdf

───────────────────────────────

**Fig. 1.** Figure1

[Figure]

---

## Author Comment (AC2) · 5 Sep 2019

Comment #01:It is acknowledged that the ionosphere is an important and indispensable part of atmosphere for earth. Please cite some past papers (at lease one

Response #1: The following reference has be cited in the manuscript on page 1 line 28. Wu, J., Quan, K. H., Dai, K. L., et al.: Progress in the study of the Chinese reference ionosphere, Advances in Space Research, 18(6), 187-190, 1996.

[Figure]

Please also note the supplement to this comment:
https://www.ann-geophys-discuss.net/angeo-2019-72/angeo-2019-72-AC2-
supplement.pdf

———————————————————

[Figure]

**Supplement:**

**Combinatorial observation ionospheric characteristics during tropical cyclone Debbie passing eastern Australia in 2017 using GPS and ionosonde**

Fuyang Ke[1], Jinling Wang[2], Kehe Wang[3], Jiujing Xu[4], Yong Wang[5], Xinzhi Wang[1], Jian Deng[6]

[1]School of Remote Sensing and Geomatics Engineering, Nanjing University of Information Science & Technology, Nanjing 210044, China

[2]School of Civil and Environmental Engineering, The University of New South Wales, Sydney, 2025, Australia

[3]Space Weather Services, Bureau of Meteorology, Surry Hills, New South Wales, Australia

[4]School of Geodesy and Geomatics, Anhui University of Science and Technology, Huainan 232001, China

[5]Jiangsu Province Surveying and Mapping Engineering Institute, Nanjing 210013, China

[6]School of Computer and Information Engineering, Xiamen University of Technology, Xiamen 361024, China

*Correspondence to*: Fuyang Ke (ke.fuyang@qq.com)

**Abstract.** The ionospheric morphology responses to tropical cyclone passing over eastern Australia, named as DEBBIE in 2017, is investigated using Global Positioning System (GPS) Slant Total Electron Content (STEC), GPS ionospheric scintillation $S_4$ index and ionospheric characteristics by ionosonde. Based on the data analysis in this study, some significant morphological characteristics of ionospheric responses to tropical cyclone Debbie are identified as follows: a) As the GPS satellites PRN01 and PRN11 were passing above typhoon centre, their ROTI (Rate of STEC index) values are obviously increased. b) The $S_4$ intensity of the GPS ionospheric scintillations is significantly enhanced on March 27, which mostly concentrate around tropical cyclone centre and distribute over the region within 18 °S-25 °S. c) The stronger enhancement of $f_0$F1 and $f_0$F2 are observed by ionosonde at Townsville on March 28, when the distance between Townsville and the centre of tropical cyclone Debbie was shorter. Regarding the coupling mechanism between the ionospheric disturbance and the tropical cyclone, it is supposed that the electric field perturbations due to turbulent top movement from tropical cyclones might generate ionospheric irregularity and disturbance. When radio signals encounter the bubbles produced by some ionospheric irregularities, the ionospheric scintillations occur.

**1. Introduction**

It is acknowledged that the ionosphere is one important and indispensable part of atmosphere for earth and a complex physical system (Wu et al., 1996). The ionospheric irregularity and disturbance can degrade the operation reliability of ground-based radio and GPS. Even it can lead to their failure due to loss of signal lock. Normally, ionospheric disturbances are majorly caused by the strong solar and geomagnetic activity. It has been indicated that ionospheric dynamics behaviours at low, mid and high latitude are dominated by solar tides and horizontal geomagnetic field lines, the inner magnetosphere and neutral winds, and the solar wind and electron precipitation, respectively (Skone et al., 2001). The lower layer of the ionosphere is also connected to the neutral atmosphere. In the 1950s, Beynon et al. (1953) have pointed out that ionospheric disturbance is related to the atmospheric activity in the troposphere. Moreover, the lower-level atmospheric activity not only can cause the ionospheric structure and physical variation, but also can trigger the small and medium scale ionospheric disturbances (Forbes, 1996).

The tropical cyclone is a typical and strong convective atmospheric activity. Bauer (1958) has firstly discovered one phenomenon of ionospheric response to hurricane passage that $f_0$F2 and the critical frequency of the ionospheric F2 layer observed by ionosonde began to increase as hurricane approaching to ionosonde observation station. Additionally, the total

40 electron content (TEC) and ionospheric scintillation can be achieved by ground GPS station to study the morphological characteristic of ionospheric response to tropical cyclone (Yang and Liu, 2016). Some previous studies have shown that the anomalies of TEC increment were detected during typhoon Matsa in 2005 (Mao et al., 2010), Aili in 2004 (Cheng et al., 2013), Nakri in 2008 (Lin, 2012), Tembin in 2012 (Yang and Liu, 2016) and Typhoon Meranti in 2016 (Chou et al., 2017). However, it is contrarily demonstrated that the TEC around the equatorial area decreases during typhoon Mahasen in 2013

45 and Hudhud in 2014 passing over the Indian sector (Guha et al., 2016). Meanwhile, the previous results have shown that the ionospheric disturbances are emanating outward and lasting for more than 10 h before Super Typhoon Meranti in 2016 landfall (Chou et al., 2017), and that the number of radio occultation scintillation increases to the peak as typhoon Tembin in 2012 closest to Hong Kong (Yang and Liu, 2016). The results of statistical analysis have shown that ionospheric disturbance percentage of 24 strong typhoons in China (Xiao et al., 2007), 41 tropical cyclones in the Atlantic Ocean (Nina et al., 2017),

50 and 25 hurricanes in western and central part of the Czech Republic (Šindelářová et al., 2009) are 92%, 88% and 8%, respectively. Furthermore, the characteristics of ionospheric disturbance intense are not the same. Although the results (Xiao et al., 2007; Rice et al., 2012; Yu et al., 2010; Ke et al., 2018) show that the $f_0F2$ values increase during the studied tropical cyclones, the studies (Liu et al., 2006; Rozhnoi et al., 2014) demonstrate that they decrease. Using the ionosonde instruments on board the Cosmos 1809 satellite (Isaev et al., 2010), it is indicated that the pressure of the electron gas,

55 electric field and scintillation intense increase in some specific zones. But plasma density and pressure above typhoon eye sharply decrease along with typhoon intensification.

In summary, there are still some uncertainty about morphological characteristics of ionospheric disturbances caused by tropical cyclones (Perevalova et al., 2011; Zakharov & Kunitsyn, 2012). Moreover, the ionospheric response to tropical cyclones in the southern hemisphere are rarely studied. Therefore, the previous studies are still limited not only in

60 ionospheric observation instrument but also in the representative of the tropical cyclone cases studied worldwide. In the southern hemisphere, there are several ionosonde and GPS continuously operating reference stations (CORS) distributed around Australia for detecting the ionospheric morphological parameters. In March 2017, tropical cyclone Debbie is the strongest tropical cyclone in the Australian region since tropical cyclone Quang in 2015, which is branded the most dangerous cyclone to impact Queensland since tropical cyclone Yasi in 2011. The combination observation of ionosonde,

65 GPS STEC and GPS ionospheric scintillation will be utilized to study the morphological characteristics of ionospheric response to tropical cyclone Debbie as a representative case in the southern hemisphere.

Additionally, there are also some controversies about the coupling mechanism between tropical cyclones and ionospheric disturbance. Hung et al. (1978) of NASA in the United States have found the existence of gravity waves and mesoscale disturbances in the F layer of the ionosphere during the tornado eruption using ionosonde, and considered that the gravity

70 wave may be the main source of ionospheric disturbance. It is suggested (Shen, 1982; Liu, et al. 2006; Wang, et al. 2005) that the turbopause motion is a possible mechanism for the interaction between the lower layers of the atmosphere and ionosphere. It is considered that the electric field disturbance in typhoon or hurricane is activated by the ionospheric disturbance caused by the current perturbation from the charged water droplets and aerosols transmitted upward (Isaev 2002, 2010). So, the study on tropical cyclone Debbie is also valuable to realize the coupling mechanism between ionospheric

75 disturbances and tropical cyclones.

In the following section, the using dataset and methodology are firstly introduced. Then, the ionospheric response to tropical cyclone Debbie in 2017 will be analysed for demonstrating the morphological characteristics of ionospheric response to tropical cyclone Debbie in the southern hemisphere. Meanwhile, the possible coupling mechanism of reaction between ionospheric disturbance and tropical cyclone will be discussed.

**2. Dataset**

**2.1 Tropical cyclone Debbie and ionospheric dataset**

[Figure]

**Figure 1:GPS station of Ionospheric Scintillation Monitors (Red triangle: Willi Island), International GNSS Station (Pink triangle: TOW2), Ionosonde Stations (Blue triangles: Learmonth, Townsville, Brisbane), the path of tropical cyclone DEBBIE (Red line), the tropical cyclone moving directions (arrow) and the place with the largest wind velocity (Red pentagram)**

Tropical cyclone Debbie is formed above the south of Solomon Sea of South Pacific at Universal Time (UT) 12:00 of 21 Mar 2017. Then it lands on Hook Island of Queensland at UT00:00 on 28 Mar 2017 at the speed of 105 knots(about 54 m/s). Tropical cyclone Debbie left the Australian continent from Brisbane at 12:00 on 30 Mar 2017. As shown in Figure 1, tropical cyclone Debbie moves from north to south and its impact zone is in medium geomagnetic latitude from $15°$S to $60°$S in the southern hemisphere. Tropical cyclone Debbie centre passes above Willi GPS ionospheric scintillation station, Townsville and Brisbane Ionosonde stations, TOW2 GPS stations. When tropical cyclone Debbie lands on Hook Island of Queensland, the ellipsoidal distances between the tropical cyclone centre and the observation stations of Willis, Townsville and Brisbane are 460 km, 230 km and 925 km, respectively.

**2.2 Solar and geomagnetic field activity**

[Figure]

**Figure 2:The solar index F10.7, the Dst and Kp indexes of geomagnetic field from16 March to 5 April 2017**

The ionospheric activity is mainly dominated and affected by the solar and geomagnetic activities. Hence, the influence from the solar and geomagnetic field should be firstly analysed before studying the ionospheric response to tropical cyclone. Normally, the solar radio flux (10.7cm/2800MHz, F10.7), Dst and Kp geomagnetic activity indexes are used to judge the solar and geomagnetic activity level, respectively. Their variations during tropical cyclone Debbie are shown in Figure 2.

Generally, the F10.7 ranges of [70sfu, 100sfu], [100sfu, 150sfu], and [150sfu, 250sfu] represent low, moderate, and high level of solar activity, respectively [Wang et al., 2015]. The Dst ranges of $-50\,\text{nT} < \text{Dst} \leq -30\,\text{nT}$, $-100\,\text{nT} < \text{Dst} \leq -50\,\text{nT}$, $-200\,\text{nT} < \text{Dst} \leq -100\,\text{nT}$ and $\text{Dst} \leq -200\,\text{nT}$ signify small, moderate, large and stronger and severe geomagnetic storms, respectively [Mao et al., 2010]. The Kp index ranges of 0-1, 2-4, 5, 6, and 7-9 denote quiet geomagnetic field, unstable geomagnetic field, small geomagnetic storm, large geomagnetic storm and severe geomagnetic storm, respectively. It is shown that the solar activity during tropical cyclone Debbie is at low level with F10.7 less than 100 sfu. But the Dst and Kp indexes indicate there are some a small geomagnetic storm on 27 March 2017 before cyclone Debbie landed on Hook Island of Queensland at UT00:00 on 28 Mar 2017. Therefore, the influence on ionosphere from the small geomagnetic storm should not be ignored to study the ionospheric response to tropical cyclone Debbie.

**3 Methodology**

**2.1 The Rate of GPS STEC**

The rate of GPS slant TEC (ROT) is as a measurement of GPS phase fluctuation, which can be used to monitor the ionospheric irregularity and disturbance. It represents the derivative of GPS slant TEC between two successive epochs, which can be calculated by the following equation (Yang and Liu, 2016):

$$\text{ROT} = \frac{\text{STEC}_k^i - \text{STEC}_{k-1}^i}{t_k - t_{k-1}} \tag{1}$$

where $i$ is the number of a GPS satellite; $t_k$ is an epoch time; the unit of ROT is TECU/min.

Rate of STEC index (ROTI) indicates the extent of the GPS phase fluctuations, which can be used to detect the occurrence of the ionospheric irregularities by the sharp TEC gradient. The ROTI represents the standard deviation of the ROT in a specific time interval (Yang and Liu, 2016):

$$\text{ROTI} = \sqrt{\langle \text{ROT}^2 \rangle - \langle \text{ROT} \rangle^2} \tag{2}$$

where the angle brackets denote the average value in a 5-min observation time interval.

**2.2 GPS ionospheric Scintillation**

The intense of GPS ionospheric scintillation is typically quantified by the $S_4$ index, which is calculated using the following equation (Kintner et al., 2007):

$$S_4 = \sqrt{\frac{\langle I^2 \rangle - \langle I \rangle^2}{\langle I \rangle^2}} - \sqrt{\frac{100}{\frac{\overline{C}}{N_0}} \left[ 1 + \frac{500}{19\frac{\overline{C}}{N_0}} \right]} \tag{3}$$

where $I$ is the intense of GPS signal, which is output from the GPS receiver tracking loop. $\overline{C/N_0}$ is the average Signal-to-Noise Ratio of satellite L1 Band in an observation period. If $S_4$ is more than 0.2, it indicates strong GPS ionospheric scintillation (Muella et al., 2008).

**2.3 Ionospheric characteristics by Ionosonde**

The ionospheric parameters of $f_0$E, $f_0$F1, and $f_0$F2 can be used to analyse the ionospheric response to cyclone in vertical direction. Accordingly, the electron density $N_e$ in each layer can be calculated using the frequency of reflection radio wave by the follow equation:

$$N_e = 1.24 \times 10^{10} f^2 \tag{4}$$

where $f$ is the critical frequency $f_0E$, $f_0F1$ and $f_0F2$ for the corresponding plasma with electron density $N_e$. The units of $f$ and $N_e$ are MHz and $m^{-3}$, respectively. The electron density $N_e$ is positively linear dependent on $f^2$. Hence, the parameters of $f_0E$, $f_0F1$, $f_0F2$ can reflect the electron density variation of ionosphere in vertical layers in response to tropical cyclone.

**4. Analysis Result and Discussion**

**4.1 GPS STEC response to tropical cyclone Debbie**

[Figure]

**Figure 3: The variation of ROT for GPS PRN23, PRN01 and PRN11 and ROTI (5min) of all GPS satellites during March 26-29 over the TOW2 IGS station.**

The GPS STEC response to tropical cyclone Debbie passing from 26-29 March 2017 is firstly analysed. The GPS STEC is extracted from TOW2 IGS station, which tropical cyclone Debbie passes above. The variation of ROT for PRN23, PRN01 and PRN11 and ROTI (5min) for all satellites during March 26-29 are shown in Figure 3. It shows that ROTI has an evident increment on UT 12:00 (Local Time (LT) =UT+8h) March 27, the day before tropical cyclone Debbie centre landed on Hook Island. When tropical cyclone Debbie centre landed on Hook Island of Queensland on March 28, the increment of ROTI is smaller. As shown in the top three of Figure 3, the variation of ROT for PRN01 and PRN11 are obviously increased as tropical cyclone Debbie landing. The Ionospheric Pierce Point (IPP) traces of GPS PRN01 and PRN11 satellites over TOW2 station are above the impact area of tropical cyclone Debbie on 28 March. The variation of ROT for PRN23 is not obvious, because the IPP trace of GPS PRN23 over TOW2 station is far away from the cyclone. Despite there is a small

geomagnetic storm on 27 March, GPS STEC extracted by the other GPS satellites far away from tropical cyclone Debbie are

155   not anomalous. Therefore, it can be inferred that the anomaly of ROTI and ROT extracted by GPS PRN01 and PRN11 above tropical cyclone centre on UT12:00 of 27 and 28 March are more likely triggered by cyclone Debbie.

**4.2 GPS ionospheric scintillation response**

[Figure]

**Figure 4: The GPS ionospheric scintillation S4 variations of GPS PRN23, PRN01 and PRN11 satellites during 22-29 March 2017.**
160   **The dotted red line is the threshold of the strong GPS ionospheric scintillation. The magenta vertical line denotes the time point when cyclone Debbie centre was the closest to GPS station**

Although ionospheric scintillation can degrade the GPS signal quality or even cause failure of the signal lock, GPS also provides a new tool for detecting ionospheric irregularity and scintillation. GPS scintillation amplitude index $S_4$ more than 0.2 indicates strong ionospheric scintillation. Figure 4 shows that the variation of GPS ionospheric scintillation of PRN23,
165   PRN01 and PRN11 above Willi Island GPS station during the period of 22-29 March 2017. It shows that the geomagnetic field is with a small storm in Figure 2. Theoretically, the affection on ionospheric scintillation of GPS PRN01 and PRN11 should be approximately same under the same geomagnetic storm. When the distance from tropical cyclone centre to Willis station is 370 km along with the wind speed of 54 m/s at midnight of 27 March, the number and intensity of S4>0.2 observed by GPS satellite PRN01 and PRN11 near to the tropical cyclone centre are more and larger than those at the other time in
170   Figure 4. But the S4 observed by GPS satellite PRN23 far away from the tropical cyclone centre are not obviously abnormal in Figure 4. Hence, it can be deduced that the number increment and enhance of GPS ionospheric scintillations might be triggered by tropical cyclone Debbie on March 27.

[Figure]

**Figure 5:Ionospheric pierce point traces and GPS ionospheric scintillation $S_4$ intensity of GPS PRN01 and PRN11 satellites through cyclone Debbie on March 27. The red dashed cycles indicate the area affected by cyclone Debbie. The colourful solid circles are GPS ionospheric scintillations and their intensity**

To further determine the correlation between GPS ionospheric scintillation and tropical cyclone Debbie, the spatial relations between GPS ionospheric scintillation points `pierce` into ionosphere and tropical cyclone Debbie are shown in Figure 5. It is obvious that more points of S$_4$>0.2 are mainly distributed around the tropical cyclone centre. What is more that the intensity and number of the points of S$_4$>0.2 above the area of $18°S - 25°S$ in the latitude and $150°E - 155°E$ in the longitude around tropical cyclone center (B $= 19.6°S, L = 149.8°E$) is stronger and larger than those above the other area. Normally, the occurrence of strong ionospheric scintillations is more frequent in low geomagnetic latitude ($\pm15°$) and high geomagnetic latitude ($\pm70°$). Nevertheless, the area of these GPS ionospheric scintillations with S$_4$>0.2 doesn't belong to the area of frequent ionospheric scintillation. The evidence further verified that the strong ionospheric scintillations might be triggered by tropical cyclone Debbie.

**4.3 The ionospheric parameters in E/F1/F2 layer by Ionosonde**

The ionosphere is divided into D, E, F1 and F2 layers. There are differences in the characteristics of the ionosphere in each layer. However, it is difficult to distinguish the characteristics of the ionosphere in each layer in response to tropical cyclone Debbie only used by GPS. Therefore, Ionosondes installed in Townsville and Brisbane are used to detect and analyse the characteristic parameters of $f_0$E, $f_0$F1 and $f_0$F2 in E, F1 and F2 layer for ionospheric response to cyclone Debbie. With regard to characteristic parameters $f_0$E, $f_0$F1 and $f_0$F2, the monthly median values from 24 to 31 March are referred as their normal values compared with the electron density response to tropical cyclone in vertical layers.

[Figure]

**Figure 6:The $f_0$E variation in the ionospheric E layer at Learmonth, Townsville and Brisbane Ionosonde stations from 24 to 30 March in 2017 as tropical cyclone Debbie moving. The $f_0$E is the ordinary wave critical frequency of the lowest thick layer which causes a discontinuity. The red lines denote the monthly median value of $f_0$E from 24 to 30 March**

The characteristic parameters $f_0$E in the ionospheric E layer observed by Townsville and Brisbane Ionosonde stations as tropical cyclone Debbie moving from 24 to 30 March are shown in Figure 6. In spite of $f_0$E values observed by Townsville Ionosonde are small enhanced and elevated at midday of 29 and 30 March after tropical cyclone Debbie landfall, the $f_0$E values observed by Learmonth, Townsville and Brisbane Ionosondes on landfall day and other days are not significantly anomalous compared their monthly median values. The $f_0$E observed by Learmonth, Townsville and Brisbane Ionosondes are all approximately equal. The ranges of $f_0$E and virtual height (h'E) are from 1.7 to 4.0 MHz and from 85 km to 100 km, respectively. The phenomenon in the ionospheric E layer indicates that tropical cyclone Debbie could not disturb the ionosphere in E layer.

[Figure]

**Figure 7: The $f_0$F1 variation in the ionospheric F1 layer at Learmonth, Townsville and Brisbane Ionosonde stations from 24 to 30 March in 2017 as tropical cyclone Debbie moving. The $f_0$F1 is the ordinary wave F1 critical frequency. The red lines denote the monthly median value of $f_0$F1 from 24 to 30 March**

[Figure]

**Figure 8: The $f_0$F2 variation in the ionospheric F2 layer at Learmonth, Townsville and Brisbane Ionosonde stations from 24 to 30 March in 2017 as tropical cyclone Debbie moving. The $f_0$F2 is ordinary wave critical frequency of the highest stratification in the F region. The red lines denote the monthly median value of $f_0$F2 values from 24 to 30 March. The pink rectangular is tropical cyclone Debbie landfall day**

The ionospheric parameters $f_0$F1 and $f_0$F2 on Learmonth, Townsville and Brisbane Ionosonde stations as tropical cyclone Debbie moving from 24 to 30 March are shown in Figure 7 and Figure 8. It is obvious that the $f_0$F1 and $f_0$F2 intensity on Townsville station on 28 March are significantly stronger than the monthly median value. Meanwhile, the $f_0$F2 intensity on Learmonth and Brisbane far away from tropical cyclone Debbie centre are approximately equal to the monthly median values under the same solar and geomagnetic activity. Theoretically, the influences on $f_0$F1 and $f_0$F2 of Learmonth and Townsville in the same latitude under the same small geomagnetic storm condition on 27 and 28 March should be almost the same. Nevertheless, the anomaly extent of $f_0$F1 and $f_0$F2 on Townsville station are significantly larger than those on Learmonth on 28 March. The largest $f_0$F1 deviation relative to the monthly median value on Townsville station is 1.0MHz -

1.5MHz. Additionally, the largest $f_0F2$ deviation relative to the monthly median value on Townsville station is 5.0MHz. The electron density $N_e$ is positively related to $f_0F1$ and $f_0F2$. Thus, it can be inferred that the electron density of ionospheric F2 layer significantly increased as tropical cyclone Debbie is above Townsville station. Despite the $f_0F1$ in F1 layer observed by Brisbane station is also increased on 28 March in Figure 7, the periodic anomaly of $f_0F1$ in those day might be due to Ionosonde noise. Therefore, the stronger enhancement of $f_0F1$ and $f_0F2$ observed by Townsville Ionosonde station on 28 March should be attributed to tropical cyclone Debbie.

**4.3 The mechanism of ionospheric response to tropical cyclone**

[Figure]

**Figure 9: Profile of atmosphere and tropical cyclone structure (From China Meteorological Administration http://www.cma.gov.cn) and the schematic diagram of ionospheric response to tropical cyclone**

The results reveal that the ionospheric irregularity and disturbance could be likely related to tropical cyclone Debbie. Even it can further produce GPS ionospheric scintillations. However, the coupling mechanism between tropical cyclone and ionospheric disturbances is still indefinite and controversial. The previous studies considered that the source of ionospheric disturbance might be from gravity waves generated by tropical cyclone (Xiao, et al. 2007), an atmosphere divergence/convergence model and dynamic coupling (Shen, 1982), a disturbed electric field caused by tropical cyclone (Isaev, et al. 2002), turbulent top layer movement of tropical cyclone (Shen, 1982; Liu, et al. 2006; Wang, et al. 2005) and lighting discharge from clouds of tropical cyclone (Shao, et al. 2013). Nevertheless, the vertical gravity waves could not disturb the more than 100 km height ionosphere in effective range due to its dozens of kilometres wavelength. Especially, it is much difficult to explain why $f_0F2$ in more than 300 km height F2 ionospheric layer observed by Townsville Ionosonde station is larger than the monthly median value as tropical cyclone Debbie was landing on 28 March in Figure 8. Assuming that gravity waves can cause ionospheric disturbances, the $f_0E$ in ionospheric E layer should be also anomalous. On the contrary, the $f_0E$ is approximately equal to the monthly median value in Figure 6. It is also difficult to explain the phenomenon only using the turbulent top layer movement of tropical cyclone and the disturbed electric field above the tropical cyclone. Because the tropical cyclone belongs to airflow system in troposphere with a height of 20 km or less and

might not directly affect the ionosphere with a height of 50 km or more in the left of Figure 9. Furthermore, the short-term lightning could not explain the long-term ionospheric disturbances for about 2 hours in Figure 8.

Therefore, it is supposed that the ionospheric disturbance in response to tropical cyclone Debbie is interacted by multi-source
250 of turbulent top layer movement, electric field and electron photochemical reactions. The strong tropical cyclone airflows can lead to the structure change of stratosphere and mesosphere. Among them, the upward airflow will continue to develop upward due to the temperature structure of the middle layer and elevate the turbulence layer with about 100km height (Shen, 1982). By contrary, the airflow direction of the tropopause above tropical cyclone centre is downward. According to atmospheric turbulence layer movement theory (Liu, et al. 2006), the airflows from the tropical cyclone will make the
255 turbulent diffusion coefficient increase and the molecular diffusion coefficient decrease. As a result, some neutral molecules ($N_2$, $O_2$) in E layer will be taken into the ionospheric F1 and F2 layer. X-rays and extreme ultraviolet rays from the sun can make these neutral molecules to ionize and produce free electrons and ions leading to the increment of electron density in F1 and F2 layer (Liu, et al. 2011). Therefore, it can be explained the phenomenon that the $f_0$F1 and $f_0$F2 on Townsville station are significantly enhanced as tropical cyclone Debbie is above Townsville station in Figure 7 and Figure 8. Along with the
260 increment of electrons, the balance of electric field is destroyed. The growth of Rayleigh–Taylor instability from the electric field perturbations can lead to some ionospheric irregularities in the F layer (Prakash 1999), which may have velocity shear mixing within the hole gradients (Kelley 1985). When the hole arrives at the topside of F layer, the bubble is produced. As it happens that GPS signal encounters the bubble, ionospheric scintillation will occur. Therefore, it can explain why there are just some ionospheric scintillations of PRN01 and PRN11 above tropical cyclone Debbie, whose values of $S_4$ are more than
265 0.2 in Figure 5. Moreover, the S4 of ionospheric scintillations of PRN01 and PRN11 observed by Townsville station is more than 0.2 on the midday of 27 March 2017 in Figure 4. Meanwhile, the $f_0$F2 in ionospheric F2 layer is enlarged at the same time in Figure 8. Thus it can be supposed that the ionospheric scintillation is produced by the ionospheric irregularities in F2 layer due to tropical cyclone Debbie.

**5. Conclusions**

270 The morphological characteristics of ionospheric response to tropical cyclone Debbie passing eastern Australia in 2017 is investigated by GPS and ionosonde. The results agree with the previous viewpoint that tropical cyclone can trigger ionospheric disturbance. The morphological characteristics of ionospheric response to tropical cyclone Debbie can be summarized as follows.

(1) As the GPS satellites PRN01 and PRN11 were passing over tropical cyclone Debbie, their ROTI and ROT are
275 significantly increased. (2) The $S_4$ intensity of GPS ionospheric scintillations is enhanced on March 27, which mostly concentrate above tropical cyclone centre and distribute over the region of $18°S - 25°S$ in the latitude and $150°E - 155°E$ in the longitude around tropical cyclone centre ($B = 19.6°S, L = 149.8°E$)。(3) Compared with those on Learmonth and Brisbane, the intensity of $f_0$F1 and $f_0$F2 on Townsville was obviously increased as tropical cyclone Debbie landed on 28 March. At the same time, distance between Townsville and the tropical cyclone centre is the shortest.

280 Considering the influence from the geomagnetic and solar activity, the turbulent top movement theory is utilized to explain how tropical cyclone causes ionospheric irregularity and further triggers ionospheric scintillations. It is assumed that the turbulent top movement of tropical cyclone can break the balance of the electric field. Then, the electric field perturbations can contribute to the growth of Rayleigh-Taylor instability producing ionospheric disturbance in the F layer of the ionosphere. Furthermore, the ionospheric density irregularities can form drift waves to create bubbles in F layer. When GPS
285 signals encounter the bubbles, GPS ionospheric scintillations can appear.

**Funding**

This study is supported by the National Natural Science Foundation of China (grant no. 41674036 and 41704008), Talents of Six Peaks in Jiangsu Province, QingLan Project of Jiangsua.

**Acknowledgements**

290 We acknowledge International GPS Service for GPS raw data and the Bureau of Meteorology of the Australian Government for providing us with the tropical cyclone, GPS ionospheric scintillation and ionosonde data sets. Finally, we are grateful to the US National Oceanic and Atmospheric Administration and Geomagnetic Data Centre for the F10.7, Kp and Dst data support.

---

## Referee Comment (RC2) · Anonymous Referee #2 · 5 Nov 2019

Since the authors already revised this manuscript along comments on Referee #1, the following comments are applied to the revised manuscript which the authors posted on 5th September as AC1.

In this manuscript, the authors examined the ionospheric disturbances associated with cyclone Debbie. Using GPS data (ROTI and S4 index) and ionosonde data (foE, foF1, foF2), the authors showed that, as the cyclone passed, the ionospheric disturbances were generated and these parameters varied. Even though a magnetic storm developed simultaneously in passing of the cyclone, the generation of these ionospheric

disturbances seems to be related to the cyclone. The purpose of this paper is important and suitable for the publication of the journal. However, the reviewer thinks that the major revisions are still inevitable along the following comments to publish this manuscript. Withdraw and resubmission of this manuscript are also preferable.

Major comments:

1. Even though the authors had improved the English of the manuscript, there are a number of incomprehensible expressions. The referee strongly recommended the further improvement of the English of this manuscript using professional English editing services.

2. Much more detailed informations of the track of the cyclone are important. The authors showed only the times of the landing on Hook Island and leaving from Brisbane. Since cyclone Debbie may affect ionospheric disturbances more than 2 days, the positions of the cyclone on shore are very informative because the enhancements of ROTI are appeared in the limited period. The information of this cyclone is available such as Bureau of Meteorology of Australia (http://www.bom.gov.au/cyclone/history/debbie17.shtml). In addition, wind speed and the center pressure are also informative.

3. As shown in Figure 3, the authors used STEC data derived from PRN23, PRN01, PRN11. Why don't you use STEC data derived from the other GPS satellites ? Since the authors used data derived from only 3 satellites, the variations of ROTI in daytime were not examined. In daytime, is there no STEC data showing the variations of ROTI ? ROTI data in daytime is also very useful in comparison with the ionosonde data.

4. In Figure 3, in addition to the previous comment, ROT for PRN01 and PRN11 are fluctuated but that for PRN23 is not. The authors explain that "the IPP trace of GPS PRN23 over TOW2 station is far away from the cyclone." To confirm this explaination, the traces of IPPs for PRN23 are necessary. How far the IPPs for PRN23 is from the cyclone ? On the other hand, the variations of ROT determined by PRN01 and PRN11

appeared around 12UT. This may be related to the distance between the cyclone and IPP for PRN01 and PRN11. In order to show how effective the distances between the cyclone and IPPs are for the variation of ROT, the tracks and positions of IPPs for PRN01 and PRN11 are also important.

5. The comments #3 and #4 are also applicable to S4 data. How close IPPs were close to the cyclone when the scintillations occurred ?

6. In Figure 5, the authors show a map of S4 intensity. The authors explained that "What is more that the intensity and number of the points of S4>0.2 above the area of $18°S − 25°E$ in the latitude and $150°E − 155°E$ in the longitude around tropical cyclone center (B = 19.6°S, L = 149.8°E) is stronger and larger than those above the other area." From the referee's view, the enhancement of S4 index is also appeared in the northern area (0S-18S, 145E-150E). Is not this enhancement related to the cyclone ? Even though the authors may explain this enhancement is related to the geomagnetic storm, we cannot distinguish whether this enhancement is due to the geomagnetic storm or the cyclone because the authors do not show the time of each IPP position. Basically, the enhancement of the S4 index due to geomagnetic storms appeared in wide longitudinal area. Why do not the enhancement of S4 index appear in the other longitudinal area ? More detail analysis of this data is needed.

7. In Figure 6, foE in Brisbane during 26th to 29th March was somewhat fluctuated as compared to Learmonth and Townsville. Is this fluctuation not related to the cyclone ?

8. Page 9 line 225: The authors explained that "the periodic anomaly of foF1 in those day might be due to Ionosonde noise." The small fluctuations of foE is also noise ? How about ionograms in this period ?

9. Page 9 line 239: What does "vertical gravity wave" mean ? Basically, gravity waves cannot propagate vertical direction.

10. Page 10 line 255: The authors described "some neutral molecules (N2, O2) in E

layer will be taken into the ionospheric F1 and F2 layer. " Most of the neutral molecules, such as N2, O2, distribute around the altitude of 150 km. It is possible that these molecules might be transferred to F1 region by some turbulences. However, are these molecules transferred to F2 region by any turbulences ?

11. Page 10 line 260: The authors described the generation mechanism of equatorial plasma bubbles, which cause scintillations of GPS signals. As for the source of the bubbles, the ionospheric perturbations are important. On the other hand, as for the growth rate of Rayleigh-Taylor instability, not the electric field perturbation but the eastward electric field is important. The referee wonder if perturbations always generate the eastward electric field ? In the present case, the growth rate happens to be larger ? This explanation cannot be applied to all the cases for ionospheric disturbances by cyclones.

Minor comments:

1. Caption of Figure 1 : The locations the ionosonde are shown by Blue pentagrams not Blue triangles.

2. Page 3 line 92: What is "ellipsoidal distance" ?

3. Equation (1) : The definitions of ROT and ROTI were originally submitted by Pi et al. (1997).

4. Page 6 line 168: "midnight of 27 March" is 0UT or 24UT on 27 March ?

5. The location (latitude and longitude) of the cyclone center is shown by (B, L), e.g. page 7 line 181, page 10 line 277. This expression is not familiar with those related to Aeronomy field.

6. Page 9 line 239: Shao et al. (2013) is not listed in Reference.

Reference:

Pi et al., Monitoring of global ionospheric irregularities using the Worldwide GPS Network, Geophys. Res. Lett., 24, 18, 2283, 1997.

---

## Short Comment (SC4) · 7 Nov 2019

My comment is Section 5 for Conclusions The Conclusions can be shorter.

---

## Author Comment (AC3) · 6 Dec 2019

Comment #01: Even though the authors had improved the English of the manuscript, there are a number of incomprehensible expressions. The referee strongly recommended the further improvement of the English of this manuscript using professional English editing services. Response #01: The manuscript has been edited for proper English language, grammar, punctuation, spelling, and overall style by one or more of the highly qualified native English speaking editor at AJE. The certificate is attached.

[Figure]

Comment #02: Much more detailed information of the track of the cyclone are important. The authors showed only the times of the landing on Hook Island and leaving from Brisbane. Since cyclone Debbie may affect ionospheric disturbances more than 2 days, the positions of the cyclone on shore are very informative because the enhancements of ROTI are appeared in the limited period. The information of this cyclone is available such as Bureau of Meteorology of Australia (http://www.bom.gov.au/cyclone/history/debbie17.shtml). In addition, wind speed and the centre pressure are also informative. Response #02: The more detailed information of DEBBIE cyclone have been added on page 3 line 93-98. As shown in Table 1, the maximum wind speed and minimum air pressure of Debbie cyclone centre reach 54 m/s and t 944 hPa at UT 12:00 on March 27, respectively. At the same time, the cyclone centre is about 111 km away from its landfall point. After DEBBIE cyclone landfall, the wind speed of the cyclone centre decreased rapidly. The wind speed of Debbie cyclone centre drops to 15 m/s and the air pressure increases to 1000 hPa after 24 hours. Table 1: The wind velocity and pressure of DEBBIE cyclone centre and the distance from its centre to landfall point Time Velocity (m/s) Pressure (hPa) Distance (km) 0327, 00:00 41 963 188.1 0327, 06:00 46 956 155.7 0327, 12:00 54 944 111.0 0327, 18:00 54 944 38.4 0328, 00:00 54 960 0 0328, 06:00 39 977 52.8 0328, 12:00 23 989 129.3 0328, 18:00 21 993 210.4 0329, 00:00 15 1000 267.7 The locations of the days before and after DEBBIE cyclone landfall have been added in Figure 1

Figure 1: GPS stations of ISMs (Red triangles: Willi Island), GPS stations of IGS (Pink triangles: TOW2), Ionosonde stations (Blue pentagrams: Learmonth, Townsville, Brisbane), paths of tropical cyclone DEBBIE (Red line), the tropical cyclone moving directions (arrows) and the places of cyclone centre before and after DEBBIE landfall (Blue points)

Comment #03: As shown in Figure 3, the authors used STEC data derived from PRN23, PRN01, PRN11. Why don't you use STEC data derived from the other GPS satellites? Since the authors used data derived from only 3 satellites, the variations of

ROTI in daytime were not examined. In daytime, is there no STEC data showing the variations of ROTI? ROTI data in daytime is also very useful in comparison with the ionosonde data. Response #03: The variations of ROTI (5 min) for all GPS satellites in daytime and nighttime during March 26-29 are shown in the bottom subplot of Figure 3. Compared with the other GPS satellites, the variations of ROTI observed by PRN01 and PRN11 denoted by blue and purple lines in the bottom subfigure are obviously larger at 12:00 on 27 March before DEBBIE cyclone landfall. For space limitation, only ROTI observed by PRN23 as a representative is compared with those observed by PRN01 and PRN11 in detail. The lack of ROTI derived from PRN23, PRN01 and PRN11 in daytime is due to the periodic variation of GPS satellite motion.

Figure 3: The variation of ROT for GPS PRN23, PRN01 and PRN11 and ROTI (5min) of all GPS satellites during March 26-29 over the TOW2 IGS station.

Comment #04: In Figure 3, in addition to the previous comment, ROT for PRN01 and PRN11 are fluctuated but that for PRN23 is not. The authors explain that "the IPP trace of GPS PRN23 over TOW2 station is far away from the cyclone." To confirm this explanation, the traces of IPPs for PRN23 are necessary. How far the IPPs for PRN23 is from the cyclone? On the other hand, the variations of ROT determined by PRN01 and PRN11appeared around 12UT. This may be related to the distance between the cyclone and IPP for PRN01 and PRN11. In order to show how effective, the distances between the cyclone and IPPs are for the variation of ROT, the tracks and positions of IPPs for PRN01 and PRN11 are also important. Response #04: The following figure has been added on page 6 line 150. And description of Figure 4 has been added on page 7 line 160-162.As shown in Figure 4, the Ionospheric Pierce Point (IPP) traces of GPS PRN01 and PRN11 satellites over TOW2 station are above the impact area of tropical cyclone Debbie on 28 March. The variation of ROT for PRN23 is not obvious, because the IPP trace of GPS PRN23 over TOW2 station is far away from the track of Debbie cyclone centre.

Figure 4: GPS stations of ISMs (Red triangles: Willi Island), GPS stations of IGS (Pink

triangles: TOW2), paths of tropical cyclone DEBBIE (Red line), the Ionospheric Pierce Point (IPP) trajectories and orientation of PRN01 (Blue line), PRN11 (Purple line) and PRN23 (Red line) GPS satellites on 27 March

Comment #05: The comments #3 and #4 are also applicable to S4 data. How close IPPs were close to the cyclone when the scintillations occurred? Response #05: According to the suggestions of comments #3 and #4, the variations of S4 for all GPS satellites in daytime and nighttime during March 26-29 are added in the bottom subplot of Figure 5. The IPP trajectories and orientation of GPS PRN01, PRN11 and PRN23 satellites are shown in Figure 4. It is obvious that the S4 of GPS PRN01 and PRN11 is stronger than that of the other GPS satellites at about UTC 12:00 on 27 March in Figure 5.

Figure 5: The GPS ionospheric scintillation S4 variations of GPS PRN23, PRN01, PRN11 and PRN01-32 satellites during 22-29 March 2017. The dotted red line is the threshold of the strong GPS ionospheric scintillation. The magenta vertical line denotes the time point when cyclone Debbie centre was the closest to GPS station The distances from IPPs to Debbie cyclone centre are described as following. When the distance from tropical cyclone centre to Willis station is 370 km along with the wind speed of 54 m/s at 12:00 of 27 March, the number and intensity of S4>0.2 observed by GPS satellite PRN01 and PRN11 near to the tropical cyclone centre are larger and stronger than those of the other GPS satellites in Figure 5. At the same time point, the distances from DEBBIE cyclone centre to PRN01 and PRN11 are 300 km and 453 km, respectively. Comment #06: In Figure 5, the authors show a map of S4 intensity. The authors explained that "What is more that the intensity and number of the points of S4>0.2 above the area of 18âŮęS − 25âŮęE in the latitude and 150âŮęE − 155âŮęE in the longitude around tropical cyclone centre (B = 19.6âŮęS, L = 149.8âŮęE) is stronger and larger than those above the other area." From the referee's view, the enhancement of S4 index is also appeared in the northern area (0S-18S, 145E-150E). Is not this enhancement related to the cyclone? Even though the authors may explain
this enhancement is related to the geomagnetic storm, we cannot distinguish whether this enhancement is due to the geomagnetic storm or the cyclone because the authors do not show the time of each IPP position. Basically, the enhancement of the S4 index due to geomagnetic storms appeared in wide longitudinal area. Why does not the enhancement of S4 index appear in the other longitudinal area? More detail analysis of this data is needed. Response #06:

Figure 6: Ionospheric pierce point traces and S4 intensity of GPS ionospheric scintillation observed by PRN01, PRN11 and PRN23 satellites through cyclone Debbie on March 27. The red line indicates the path of cyclone Debbie centre. The colourful solid circles are GPS ionospheric scintillations and their intensity According to your suggestions, the S4 values of PRN01, PRN11 near Debbie centre and PRN23 far from Debbie centre on 27 March are compared. The IPP traces and S4 intensity of GPS ionospheric scintillation observed by PRN01, PRN11 and PRN23 satellites on 27 March has been redrawn and shown in Figure 6. The colourful solid circles represent the IPPs of GPS ionospheric scintillations and their intensity. It is obvious that the intensity of GPS ionospheric scintillation for PRN01 and PRN11 with the path of the Debbie cyclone centre is significantly stronger than that of RPN23 further away from Debbie. The IPPs with stronger GPS ionospheric scintillations are mainly distributed around the outer edge of Debbie. Under the same geomagnetic conditions, the intensity of the GPS ionospheric scintillations for all GPS satellites should be approximately identical. Therefore, the difference in the intensity of GPS ionospheric scintillation for PRN01, PRN11 and PRN23 further verifies that Debbie might have enhanced the intensity of ionospheric scintillation.

Comment #07: In Figure 6, fo'E in Brisbane during 26th to 29th March was somewhat fluctuated as compared to Learmonth and Townsville. Is this fluctuation not related to the cyclone? Response #07: The f0E observed by Learmonth and Townsville Ionosonde are all approximately equal. In Figure 7, f0E in Brisbane during 26th to 29th March is somewhat fluctuated as compared to Learmonth and Townsville. But the

ionosonde instrument at Townsville closer to DEBBIE cyclone centre did not observe similar small disturbances. Hence, the small fluctuation of f0E in Brisbane during the period from 26th to 29th March could not be disturbed by cyclone Debbie. The reason is that the distances from Brisbane station to DEBBIE cyclone centre are more far on those days.

Comment #08: Page 9 line 225: The authors explained that "the periodic anomaly of foF1 in those day might be due to Ionosonde noise." The small fluctuations of fo'E is also noise? How about ionograms in this period ? Response #08: The ionograms at UT06:00 from 25 to 30 March above Brisbane station are shown in the following Figure. The disturbances in the ionograms on 27 March should be attributed to the geomagnetic storm shown in Figure 2, because the Debbie cyclone centre is still more far away from Brisbane station during the same period. The ionograms in F1 layer on these days are also anomalous and agreed with the fluctuations of f0F1. The source of ionospheric disturbance is complex. Therefore, the periodic anomaly of f0F1 and small fluctuations of f0E should be not directly ascribed to Ionosonde noise.

Comment #09: Page 9 line 239: What does "vertical gravity wave" mean? Basically, gravity waves cannot propagate vertical direction. Response #09: Inertio-gravity waves (IGWs) caused by typhoon can spread in vertical direction. IGWs in the stratosphere generated by Rusa have a vertical wavelength of 3–11 km (Kim, el al., 2005). Gravity wave can affect the atmospheric layer with high altitudes because of their relatively large vertical wavelengths and about 50 m/s vertical velocity (Kong, el al., 2017). [1] Kim, S. Y., et al. (2005). "A numerical study of gravity waves induced by convection associated with Typhoon Rusa." Geophysical Research Letters 32(24). [2] Kong, J., et al. (2017). "A clear link connecting the troposphere and ionosphere: ionospheric responses to the 2015 Typhoon Dujuan." Journal of Geodesy 91(9): 1087-1097.

Comment #10: Page 10 line 255: The authors described "some neutral molecules (N2, O2) in E layer will be taken into the ionospheric F1 and F2 layer. "Most of the neutral molecules, such as N2, O2, distribute around the altitude of 150 km. It is

possible that these molecules might be transferred to F1 region by some turbulences. However, are these molecules transferred to F2 region by any turbulences? Response #10: The neutral molecules in E layer is difficult to be taken into F2 layer. The neutral molecules in E layer are mainly taken into F1 layer and change the structure of F1 layer. Furthermore, the electric ions of F1 layer will be taken into F2 layer.

Comment #11: Fig. 5.: Page 10 line 260: The authors described the generation mechanism of equatorial plasma bubbles, which cause scintillations of GPS signals. As for the source of the bubbles, the ionospheric perturbations are important. On the other hand, as for the growth rate of Rayleigh-Taylor instability, not the electric field perturbation but the eastward electric field is important. The referee wonder if perturbations always generate the eastward electric field? In the present case, the growth rate happens to be larger? This explanation cannot be applied to all the cases for ionospheric disturbances by cyclones. Response #11: The production of the plasma bubbles initiated by gravity waves takes a much shorter time than that resulting from two-dimensional initial density perturbations. The Rayleigh-Taylor instability initiated by gravity waves can also produce a steep gradient on the west wall, which provides a favorable condition for excitation of smaller-scale secondary instabilities. Although the viewpoint proposed in this paper cannot be applied to the all cases for ionospheric disturbances by cyclones, it can provide a new idea for the uncertain mechanism of ionospheric disturbance caused by cyclones. The following description has been added on page 12 line 275-279. The production of the plasma bubbles initiated by gravity waves takes a much shorter time than that resulting from two-dimensional initial density perturbations. The Rayleigh-Taylor instability initiated by gravity waves can also produce a steep gradient on the west wall, which provides a favorable condition for excitation of smaller-scale secondary instabilities. When the hole arrives at the topside of F layer, the bubble is produced.

Minor comments: M1. Caption of Figure 1: The locations the ionosonde are shown by Blue pentagrams not Blue triangles. Response to M1: The mistake has been modified

on page 3 line 84

M2. Page 3 line 92: What is "ellipsoidal distance"? Response to M2: The distance between two points on the surface of the ellipsoid.

M3. Equation (1) : The definitions of ROT and ROTI were originally submitted by Pi et al. (1997). Response to M3: It has been modified on page 4 line 119 and page 13 line 356.

M4. Page 6 line 168: "midnight of 27 March" is 0UT or 24UT on 27 March? Response to M4: It has been modified on page 8 line 178.

M5. The location (latitude and longitude) of the cyclone centre is shown by (B, L), e.g.page 7 line 181, page 10 line 277. This expression is not familiar with those related to Aeronomy field. Response to M5: This has been modified on page 8 line 191 and page 12 line 294.

M6. Page 9 line 239: Shao et al. (2013) is not listed in Reference. Response to M6: The reference is added in the revised manuscript and as following. Shao, X. M., Lay, E. H., Jacobson, A. R.: Reduction of electron density in the night-time lower ionosphere in response to a thunderstorm, nature geoscience, 6, 29-33, doi: 10.1038/NGEO1668, 2013.

Please also note the supplement to this comment:
https://www.ann-geophys-discuss.net/angeo-2019-72/angeo-2019-72-AC3-supplement.zip

[Figure]

Legend:
- ▲ GPS Scintillaton (red triangle)
- ● Cyclone Center (blue circle)
- ▼ IGS GPS (magenta triangle)
- — Cyclone Track (red line)
- ★ Ionosonde (blue star)

Map labels: Learmonth, TOW2, Townsville, 0:00 29 March, Willi Island, 0:00 27 March, Brisbane

Axes: Longitude (°) from 100 to 180, Latitude (°) from -50 to 10

**Fig. 1.** Figure1

[Figure]

**Fig. 2.** Figure3

[Figure]

[Figure]

**Fig. 3.** Figure4

**Fig. 4.** Figure5

[Figure]

**Fig. 5.** Figure6

**Fig. 6.** Ionogram